# Microglial dopamine receptor elimination defines sex-specific nucleus accumbens development and social behavior in adolescent rats

Ashley M. Kopec[1,2,3], Caroline J. Smith 🄳 [1,2], Nathan R. Ayre[1,2,3], Sean C. Sweat[3,4] & Staci D. Bilbo[1,2,3]

Adolescence is a developmental period in which the mesolimbic dopaminergic "reward" circuitry of the brain, including the nucleus accumbens (NAc), undergoes significant plasticity. Dopamine D1 receptors (D1rs) in the NAc are critical for social behavior, but how these receptors are regulated during adolescence is not well understood. In this report, we demonstrate that microglia and complement-mediated phagocytic activity shapes NAc development by eliminating D1rs in male, but not female rats, during adolescence. Moreover, immune-mediated elimination of D1rs is required for natural developmental changes in male social play behavior. These data demonstrate for the first time that microglia and complement-mediated immune signaling (i) participate in adolescent brain development in a sex-specific manner, and (ii) are causally implicated in developmental changes in behavior. These data have broad implications for understanding the adolescent critical period of development, the molecular mechanisms underlying social behavior, and sex differences in brain structure and function.

[1] Department of Pediatrics, Harvard Medical School, Boston 02129 MA, USA. [2] Lurie Center for Autism, Massachusetts General Hospital for Children, Lexington 02129 MA, USA. [3] Department of Psychology and Neuroscience, Duke University, Durham 27708 NC, USA. [4] National Institutes of Mental Health, Bethesda 20892, USA. Correspondence and requests for materials should be addressed to A.M.K. (email: akopec@partners.org)

A dolescence is a developmental stage characterized by increased exploration, risk taking, and social interaction[1,2]. During this period, the mesolimbic dopaminergic 'reward' circuitry of the brain, which includes the ventral tegmental area (VTA), prefrontal cortex (PFC), and nucleus accumbens (NAc), undergoes significant developmental plasticity and neural circuit maturation[3]. A behavioral hallmark of adolescence is increased peer-centered social interaction[2]. Interestingly, access to social experience can be used in lieu of addictive drugs to induce conditioned place preference, operant conditioning, or maze performance[4], and social interactions can modulate addiction-like behaviors, acting to enhance or diminish drug-seeking behaviors in different contexts[5]. Furthermore, NAc dopamine transients are induced by brief social interactions in both adult and adolescent rats; however, adult dopamine responses habituate, while adolescent dopamine responses persist in subsequent peer interactions[6]. Taken together, these data suggest that social interaction itself is a highly rewarding/motivating experience that engages the dopaminergic reward circuitry, perhaps especially during adolescence.

Dopamine receptors in the NAc are required for social play behavior in adolescent rats[7], and two seminal studies have used optical and genetic tools to define a hypothalamic-VTA-NAc circuit underlying social behavior. Oxytocin innervation from the paraventricular nucleus of the hypothalamus (PVN) to the VTA increases VTA neuron excitability, and initiates dopamine release into the NAc, promoting pro-social behavior[8]. Remarkably, the VTA-NAc dopaminergic response was mediated specifically by NAc dopamine D1 receptors (D1rs), and not dopamine D2 receptors (D2rs), in adult female mice[9]. Interestingly, several subtypes of dopamine receptors, including D1rs, are downregulated (via unknown mechanisms) during adolescence in the PFC and dorsal striatum[10–14]. Within the NAc, there are conflicting reports: some conclude dopamine receptors are not downregulated in the NAc in adolescence[10], while other reports conclude that they are, indeed, downregulated[11,13]. Thus, the data collectively suggest that NAc D1rs are critical mediators of social behavior, but how the dopamine system in the NAc develops during adolescence, and how this development modulates social behaviors remains unclear.

Microglia, the resident immune cells of the brain, are critical mediators of neural circuit development in certain brain regions, e.g., in synaptic pruning and refinement[15]. For instance, in the developing retinogeniculate nucleus of male mice, the classical immune complement system, including complement protein C3, "tags" synapses for elimination[16]. Microglia, which exclusively express C3 receptor (C3R, often referred to as CD11b) recognize this tag, phagocytose, and lysosomally degrade the synapse. Altogether, this process results in the developmental refinement of ocular inputs[16,17]. Moreover, genetic models lacking either components of the complement signaling family or microglial-specific signaling exhibit increased synaptic numbers[15,16,18]. Remarkably, time-lapse imaging in developing organotypic hippocampal cultures has recently revealed that the role microglia play in synaptic remodeling is far more diverse than previously appreciated, and includes pre-synaptic "trogocytosis", or "nibbing", and induction of post-synaptic filopodia and post-synaptic reorganization[19]. While the complement system has been implicated through genetic manipulations in the development of the retinogeniculate nucleus[16,20] and in pathological states[21–24], it was not necessary for pre-synaptic trogocytosis in organotypic cultures of developing hippocampus[19]. Microglia/immune-mediated synaptic pruning is well characterized in early postnatal development, though its role in adolescent development has yet to be explored. Interestingly, a recent report showed increased association of synaptic elements (post-synaptic spines and

pre-synaptic glutamatergic terminals) with microglia in the prefrontal cortex during adolescence, indicating that immune-mediated synaptic pruning could also be occurring in regions with protracted development[25]. Taken together these data collectively suggest that microglia and immune signaling may constitute a ubiquitous developmental mechanism by which neural circuits, and theoretically behaviors mediated by these circuits, mature. We thus sought to determine if microglia and the complement system participate in dopaminergic development via D1r elimination in the NAc during adolescence. Furthermore, we aimed to determine whether microglia-mediated circuit refinement can account for normal developmental changes in behavior, which has not yet been established.

Herein, we report sex-specific microglia and complement-mediated phagocytosis and elimination of D1rs in the adolescent NAc. While D1rs are downregulated in both the male and female NAc, albeit at different ages, D1r downregulation is mediated via microglia and complement C3-mediated phagocytic elimination only in males. Moreover, we report for the first time that microglia-mediated receptor changes are required for natural developmental changes in behavior, specifically, adolescent social play behavior in males. Interestingly, though immune processes are not regulating D1r levels in females, perturbing the complement–microglial relationship locally in the NAc also alters social play behavior in females, suggesting that an as yet undetermined immune process is occurring in the female NAc during adolescence to mediate basal levels of social behavior. These data have broad implications for understanding the adolescent critical period of development, the molecular mechanisms underlying social behavior, and sex differences in brain structure and function.

## Results

**D1r downregulation correlates with microglial degradation.** Our first goal was to determine the pattern of D1r and C3 expression over the course of male and female adolescent development in rats. We assessed tissue at four ages: postnatal day 20 (P20) representing pre-adolescent animals, P30 representing early adolescence, P38 representing mid-adolescence, and P54 representing late adolescence. We focused our investigation on the dorso-medial area of the anterior NAc, as this area had robust D1r immunoreactivity (Supplementary Fig. 1). Importantly, throughout this report, male and female tissue was imaged with different acquisition parameters, and thus a direct comparison of males and females cannot be made. Immunofluorescent analysis revealed significant D1r downregulation in male rats between P30 and 38 (Fig. 1a; Table 1A) and in female rats between P20 and 30 (Fig. 1e; Table 1D), which was confirmed in a separate group of animals using chromogenic immunohistochemistry (Supplementary Fig. 1D,E). Next we determined if expression patterns of complement C3 correlated with D1r expression patterns. In males, C3 levels are highest at P30 (Fig. 1b; Table 1B), concurrent with peak D1r levels. However, in females, C3 levels did not peak concurrently with D1rs, and were highest at P38 (Fig. 1f; Table 1E). Interestingly, in both males and females, decreased Iba1 expression (a ubiquitous protein on microglia) was observed concurrently with D1r elimination (Fig. 1c, g; Table 1C, F).

If C3 is being utilized as a tag to recruit microglia and subsequent phagocytic activity as has been documented in the developing retinogeniculate system[16,17], then we postulated that microglia, which exclusively express C3R/CD11b, should contact C3-tagged protein complexes and mediate their elimination via lysosomal degradation. To assess this, we measured the volume of D1r and C3 signal either in contact with or inside microglia using volumetric reconstruction of Iba1 immunoreactivity to outline

microglial morphology (Fig. 1i). We used the colocalization of C3 and D1r, or the areas in which the fluorescent signals of C3 and D1r substantially overlap, as an additional tool to probe the interactions of microglia with D1rs specifically in close proximity to C3 "tags". In males, both D1rs (Fig. 1j; Table 1G) and colocalized C3-D1rs (Fig. 1l; Table 1I) are maximally contacted by microglia at P30 and/or P38, the ages between which D1rs are downregulated. Male microglial contact of C3 does not change over time (Fig. 1j; Table 1H), which was unsurprising as presumably C3 is tagging many things other than D1rs during this time. However, in females, microglia maximally contact D1rs (Fig. 1m; Table 1J) and C3 (Fig. 1n; Table 1K) at P54, rather than prior to or during D1r downregulation between P20 and 30. Interestingly, there is no significant difference in colocalized C3-

D1r contact by microglia across developmental time points (Fig. 1o; Table 1L)

Expression levels of CD68, a lysosomal protein in microglia, have been used as a marker to examine phagocytic activity, such as synaptic pruning[16,17]. Therefore, we measured the expression levels and content of CD68+ microglial lysosomes over the course of adolescent development in both sexes (Fig. 2a). In males, CD68 expression peaks from P30 to 38 (Fig. 2b; Table 2A), while in females, CD68 levels were highest at P30 (Fig. 2f; Table 2E). Interestingly, while not significant, female CD68 levels seemed to increase again at P54, raising the possibility of distinct phases of microglial phagocytic processes in the female NAc over the course of adolescence. When we examined the content of CD68+ microglial lysosomes using volumetric reconstructions

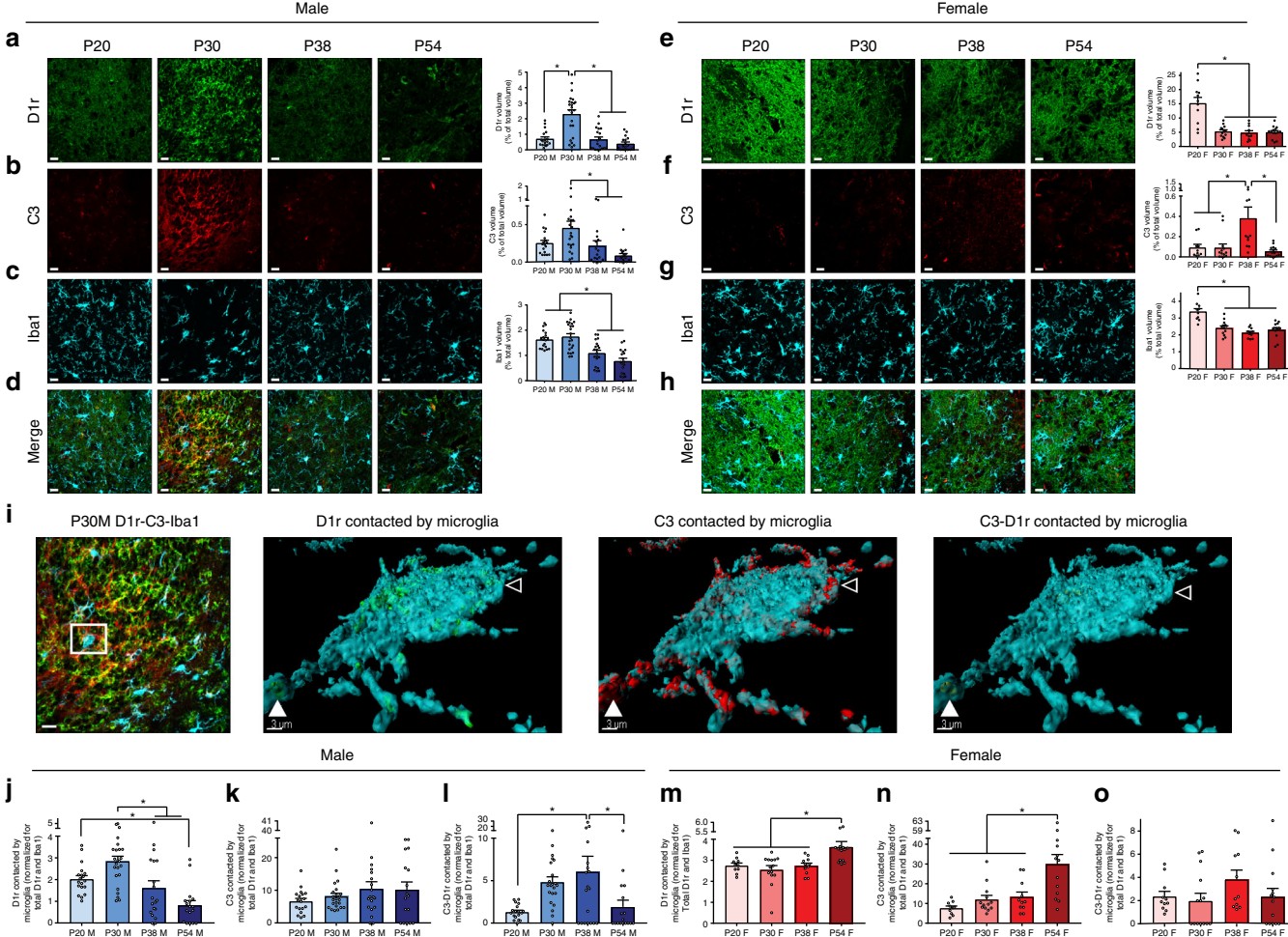

**Fig. 1** D1r downregulation in adolescence is associated with complement C3 and microglial engulfment in males, but not the females. Triple-label immunohistochemistry for D1r, C3, and Iba1 was performed and protein expression levels for each target were normalized for z-stack size. Because male and female tissue was processed at different times and required different image acquisition parameters, neither data nor images can be directly compared between the sexes. Representative images precede histograms; scale bars are equal to 20 μm. Males: **a** D1r levels transiently increased at P30 (Table 1A); **b** C3 levels decreased after P30 (Table 1B); **c** Iba1 levels decreased after P30 (Table 1C). **d** Merged male images. Females: **e** D1r levels decreased after P20 (Table 1D); **f** C3 levels transiently increased at P38 (Table 1E); **g** Iba1 levels decreased after P20 (Table 1F). **h** Merged female images. Images were then analyzed using a volumetric reconstruction of Iba1 (i.e., microglia) as a mask to determine the volume of D1r, C3, or colocalized C3-D1r physically associated with (either in contact with or inside) microglia. Masked volumes were normalized for total protein of the target and total Iba1 (see Methods). **i** Representative 2D triple-label immunohistochemistry (scale bar 20 μm), with enlarged 3D representation (scale bar 3 μm) of the selected area demonstrating (left-right) D1r, C3, and C3-D1r contacted by/internal to the microglia. Open arrow head indicates a region of the microglia where both D1r and C3 are located, but not colocalized, demonstrating the sensitivity of colocalization analyses. Closed arrow head indicates a region of the microglia where D1r and C3 colocalize (C3-D1r). Males: **j** D1rs are maximally contacted by microglia (irrespective of changing D1r and Iba1 levels) at P30 (Table 1G). **k** C3 contact by microglia does not change over time (Table 1H), and **L** C3-D1r contact is high between P30-P38 (Table 1I). Females: **m** D1rs are maximally contacted by microglia at P54, not at P20, prior to their elimination (Table 1J). **n** C3 contact is also increased at P54 (Table 1K), while **o** C3-D1r contact does not change over development (Table 1L)

**Table 1 Detailed statistics corresponding with Fig. 1**

| | Comparison | Statistical test | n | Statistic | p-value | Outliers? | Figure |
|---|---|---|---|---|---|---|---|
| A | D1r across development: Males | one-way ANOVA | 4 animals; 2 sections/animal; 1-3 zstacks/section | $F_{(3,74)}=21.51$ | <0.001 | P20: 2; P38: 1; P45: 1 | Fig. 1a |
| | P20:P30 | Holm-Sidak's posthoc | 18:24 | $t_{(74)}=5.86$ | <0.001 | | |
| | P20:P38 | Holm-Sidak's posthoc | 18:19 | $t_{(74)}=0.13$ | 0.894 | | |
| | P20:P54 | Holm-Sidak's posthoc | 18:17 | $t_{(74)}=1.04$ | 0.657 | | |
| | P30:P38 | Holm-Sidak's posthoc | 24:19 | $t_{(74)}=6.08$ | <0.001 | | |
| | P30:P54 | Holm-Sidak's posthoc | 24:17 | $t_{(74)}=6.87$ | <0.001 | | |
| | P38:P54 | Holm-Sidak's posthoc | 19:17 | $t_{(74)}=0.93$ | 0.657 | | |
| B | C3 across development: Males | one-way ANOVA | 4 animals; 2 sections/animal; 1-3 zstacks/section | $F_{(3,70)}=5.75$ | 0.001 | P20: 2; P30: 3; P38: 2 | Fig. 1b |
| | P20:P30 | Holm-Sidak's posthoc | 18:21 | $t_{(70)}=2.25$ | 0.105 | | |
| | P20:P38 | Holm-Sidak's posthoc | 18:18 | $t_{(70)}=0.41$ | 0.685 | | |
| | P20:P54 | Holm-Sidak's posthoc | 18:17 | $t_{(70)}=1.76$ | 0.230 | | |
| | P30:P38 | Holm-Sidak's posthoc | 21:18 | $t_{(70)}=2.68$ | 0.045 | | |
| | P30:P54 | Holm-Sidak's posthoc | 21:17 | $t_{(70)}=4.04$ | 0.001 | | |
| | P38:P54 | Holm-Sidak's posthoc | 18:17 | $t_{(70)}=1.35$ | 0.328 | | |
| C | Iba1 across development: Males | one-way ANOVA | 4 animals; 2 sections/animal; 1-3 zstacks/section | $F_{(3,76)}=18.57$ | <0.001 | P20: 1; P30: 1 | Fig. 1c |
| | P20:P30 | Holm-Sidak's posthoc | 19:23 | $t_{(76)}=0.83$ | 0.410 | | |
| | P20:P38 | Holm-Sidak's posthoc | 19:20 | $t_{(76)}=3.53$ | 0.002 | | |
| | P20:P54 | Holm-Sidak's posthoc | 19:18 | $t_{(76)}=5.50$ | <0.001 | | |
| | P30:P38 | Holm-Sidak's posthoc | 23:20 | $t_{(76)}=4.54$ | <0.001 | | |
| | P30:P54 | Holm-Sidak's posthoc | 23:18 | $t_{(76)}=6.57$ | <0.001 | | |
| | P38:P54 | Holm-Sidak's posthoc | 20:18 | $t_{(76)}=2.09$ | 0.078 | | |
| D | D1r across development: Females | one-way ANOVA | 4 animals; 2 sections/animal; 1-3 zstacks/section | $F_{(3,45)}=21.1$ | <0.001 | P30: 1; P38: 1; P54: 1 | Fig. 1e |
| | P20:P30 | Holm-Sidak's posthoc | 11:13 | $t_{(45)}=6.48$ | <0.001 | | |
| | P20:P38 | Holm-Sidak's posthoc | 11:11 | $t_{(45)}=6.54$ | <0.001 | | |
| | P20:P54 | Holm-Sidak's posthoc | 11:14 | $t_{(45)}=6.78$ | <0.001 | | |
| | P30:P38 | Holm-Sidak's posthoc | 13:11 | $t_{(45)}=0.33$ | 0.983 | | |
| | P30:P54 | Holm-Sidak's posthoc | 13:14 | $t_{(45)}=0.20$ | 0.983 | | |
| | P38:P54 | Holm-Sidak's posthoc | 11:14 | $t_{(45)}=0.15$ | 0.983 | | |
| E | C3 across development: Females | one-way ANOVA | 4 animals; 2 sections/animal; 1-3 zstacks/section | $F_{(3,44)}=6.97$ | 0.001 | P20: 1; P30: 1; P38: 1; P54: 1 | Fig. 1f |
| | P20:P30 | Holm-Sidak's posthoc | 10:13 | $t_{(44)}=0.03$ | 0.979 | | |
| | P20:P38 | Holm-Sidak's posthoc | 10:11 | $t_{(44)}=3.39$ | 0.006 | | |
| | P20:P54 | Holm-Sidak's posthoc | 10:14 | $t_{(44)}=0.48$ | 0.950 | | |
| | P30:P38 | Holm-Sidak's posthoc | 13:11 | $t_{(44)}=3.64$ | 0.004 | | |
| | P30:P54 | Holm-Sidak's posthoc | 13:14 | $t_{(44)}=0.48$ | 0.950 | | |
| | P38:P54 | Holm-Sidak's posthoc | 11:14 | $t_{(44)}=4.16$ | 0.001 | | |
| F | Iba1 across development: Females | one-way ANOVA | 4 animals; 2 sections/animal; 1-3 zstacks/section | $F_{(3,44)}=17.54$ | <0.001 | P20: 1; P30: 1; P38: 1; P54: 1 | Fig. 1g |
| | P20:P30 | Holm-Sidak's posthoc | 10:13 | $t_{(44)}=5.30$ | <0.001 | | |
| | P20:P38 | Holm-Sidak's posthoc | 10:11 | $t_{(44)}=6.60$ | <0.001 | | |
| | P20:P54 | Holm-Sidak's posthoc | 10:14 | $t_{(44)}=6.01$ | <0.001 | | |
| | P30:P38 | Holm-Sidak's posthoc | 13:11 | $t_{(44)}=1.61$ | 0.308 | | |
| | P30:P54 | Holm-Sidak's posthoc | 13:14 | $t_{(44)}=0.68$ | 0.551 | | |
| | P38:P54 | Holm-Sidak's posthoc | 11:14 | $t_{(44)}=0.99$ | 0.551 | | |
| G | Contacted D1r across development: Males | one-way ANOVA | 4 animals; 2 sections/animal; 1-3 zstacks/section | $F_{(3,75)}=12.34$ | <0.001 | P20: 1; P38: 1 | Fig. 1j |
| | P20:P30 | Holm-Sidak's posthoc | 19:24 | $t_{(75)}=2.47$ | 0.046 | | |
| | P20:P38 | Holm-Sidak's posthoc | 19:19 | $t_{(75)}=1.18$ | 0.244 | | |
| | P20:P54 | Holm-Sidak's posthoc | 19:17 | $t_{(75)}=3.35$ | 0.005 | | |
| | P30:P38 | Holm-Sidak's posthoc | 24:19 | $t_{(75)}=3.71$ | 0.002 | | |
| | P30:P54 | Holm-Sidak's posthoc | 24:17 | $t_{(75)}=5.92$ | <0.001 | | |
| | P38:P54 | Holm-Sidak's posthoc | 19:17 | $t_{(75)}=2.21$ | 0.060 | | |
| H | Contacted C3 across development: Males | one-way ANOVA | 4 animals; 2 sections/animal; 1-3 zstacks/section | $F_{(3,73)}=1.17$ | 0.327 | P20: 2; P30: 1; P38: 1; P54: 1 | Fig. 1k |
| I | Contacted C3-D1r across development: Males | one-way ANOVA | 4 animals; 2 sections/animal; 1-3 zstacks/section | $F_{(3,72)}=3.15$ | 0.005 | P20: 2; P30: 2; P38: 1; P54: 1 | Fig. 1l |
| | P20:P30 | Holm-Sidak's posthoc | 18:22 | $t_{(72)}=2.43$ | 0.069 | | |
| | P20:P38 | Holm-Sidak's posthoc | 18:19 | $t_{(72)}=3.15$ | 0.014 | | |
| | P20:P54 | Holm-Sidak's posthoc | 18:17 | $t_{(72)}=0.39$ | 0.701 | | |
| | P30:P38 | Holm-Sidak's posthoc | 22:19 | $t_{(72)}=0.85$ | 0.638 | | |

**Table 1** (continued)

| | Comparison | Statistical test | n | Statistic | p-value | Outliers? | Figure |
|---|---|---|---|---|---|---|---|
| | P30:P54 | Holm-Sidak's posthoc | 22:17 | $t_{(72)}=1.98$ | 0.146 | | |
| | P38:P54 | Holm-Sidak's posthoc | 19:17 | $t_{(72)}=2.72$ | 0.041 | | |
| J | Contacted D1r across development: Females | one-way ANOVA | 4 animals; 2 sections/animal; 1-3 zstacks/section | $F_{(3,45)}=6.93$ | 0.001 | P20: 1; P38: 1; P54: 1 | Fig. 1m |
| | P20:P30 | Holm-Sidak's posthoc | 10:14 | $t_{(45)}=0.71$ | 0.861 | | |
| | P20:P38 | Holm-Sidak's posthoc | 10:11 | $t_{(45)}=0.04$ | 0.967 | | |
| | P20:P54 | Holm-Sidak's posthoc | 10:14 | $t_{(45)}=3.13$ | 0.012 | | |
| | P30:P38 | Holm-Sidak's posthoc | 14:11 | $t_{(45)}=0.68$ | 0.861 | | |
| | P30:P54 | Holm-Sidak's posthoc | 14:14 | $t_{(45)}=4.20$ | 0.001 | | |
| | P38:P54 | Holm-Sidak's posthoc | 11:14 | $t_{(45)}=3.26$ | 0.011 | | |
| K | Contacted C3 across development: Females | one-way ANOVA | 4 animals; 2 sections/animal; 1-3 zstacks/section | $F_{(3,43)}=9.72$ | <0.001 | P30: 1; P38: 1; P54: 1 | Fig. 1n |
| | P20:P30 | Holm-Sidak's posthoc | 10:13 | $t_{(43)}=0.95$ | 0.571 | | |
| | P20:P38 | Holm-Sidak's posthoc | 10:11 | $t_{(43)}=1.19$ | 0.560 | | |
| | P20:P54 | Holm-Sidak's posthoc | 10:13 | $t_{(43)}=4.84$ | <0.001 | | |
| | P30:P38 | Holm-Sidak's posthoc | 13:11 | $t_{(43)}=0.29$ | 0.771 | | |
| | P30:P54 | Holm-Sidak's posthoc | 13:13 | $t_{(43)}=4.17$ | 0.001 | | |
| | P38:P54 | Holm-Sidak's posthoc | 11:13 | $t_{(43)}=3.70$ | 0.003 | | |
| L | Contacted C3-D1r across development: Females | one-way ANOVA | 4 animals; 2 sections/animal; 1-3 zstacks/section | $F_{(3,46)}=1.38$ | 0.260 | P38: 1; P54: 1 | Fig. 1o |

Statistical details for every analysis in Fig. 1

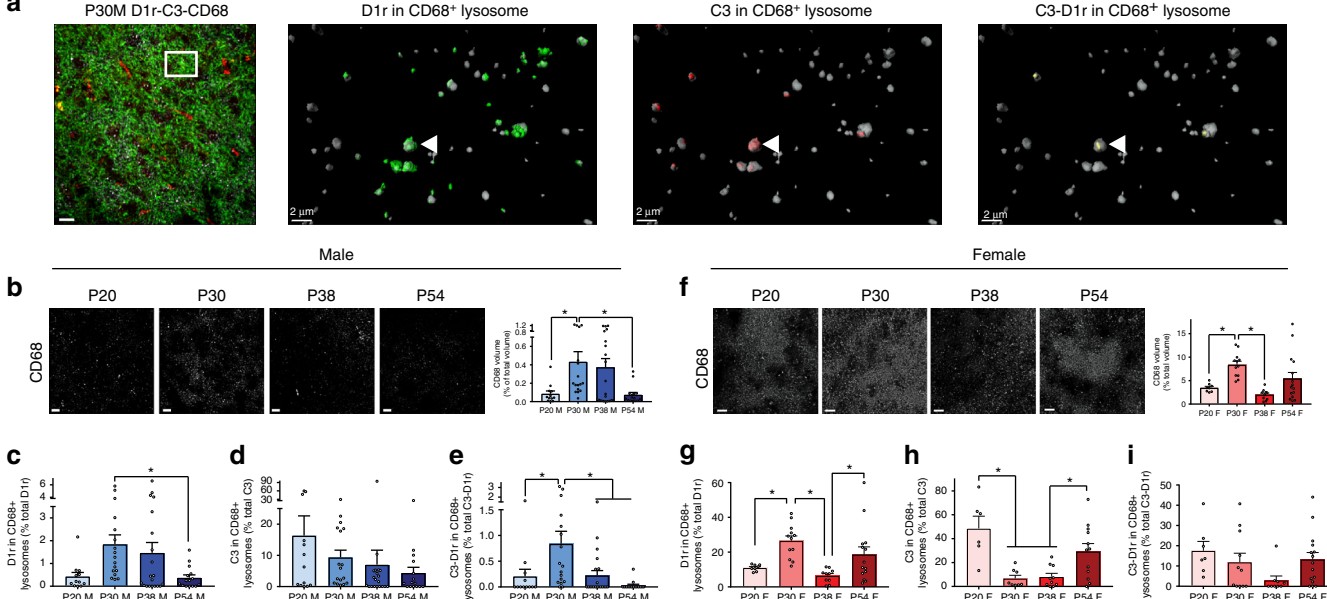

**Fig. 2** D1r downregulation in adolescence is associated with complement C3 and microglial lysosomal degradation in the male, but not the female, NAc. To assess microglial phagocytic activity and whether D1r, C3, or C3-D1r were localized in microglial CD68+ lysosomes, immunohistochemistry for CD68, D1r, and C3 was performed. **a** Representative 2D triple-label immunohistochemistry (scale bar 20 μm), with enlarged 3D representation (scale bar 2 μm) of the selected area demonstrating (left-right) D1r, C3, and C3-D1r within CD68+ lysosomes. Closed arrow head indicates a lysosome where D1r and C3 colocalize (C3-D1r). Males: **c** CD68 levels are high from P30 to 38 (Table 2A). **b** D1r content in CD68+ lysosomes is highest at P30 (Table 2B), **d** C3 content in lysosomes does not change over development (Table 2C), and **e** C3-D1r content in lysosomes is transiently elevated at P30 (Table 2D). Females: **f** CD68 levels are transiently elevated at P30 (Table 2E). **g** D1r content in CD68+ lysosomes is high at both P30 and P54 (Table 2F), **h** C3 content in lysosomes is highest at P20 and P54 (Table 2G), and **i** C3-D1r content in lysosomes does not change over development (Table 2H). For all experiments, n = 4 animals/sex/group. Data were analyzed with one-way ANOVAs and Holm-Sidak's post-hoc comparisons. Histograms portray the mean ± SEM with individual data points overlaid. Significant post-hoc Holm-Sidak t-tests (p < 0.05) comparisons are delineated with an asterisk. All statistics are in Table 2

(similar to previously described assessment of microglial contact), D1r (Fig. 2c; Table 2B) and colocalized C3-D1r (Fig. 2e; Table 2D) content within lysosomes was highest at P30 and P38 in males. Consistent with no changes in C3 contact by microglia over male adolescent development, there was no change in C3 levels associated with CD68+ lysosomes (Fig. 2d; Table 2C). In females, D1r content in CD68+ lysosomes was highest at P30 and P54, mirroring overall CD68 levels in females (Fig. 2g; Table 2F). Intriguingly, C3 lysosomal content was also bimodal during female adolescence, but with high levels at P20 and P54 (Fig. 2h;

**Table 2 Detailed statistics corresponding with Fig. 2**

| | Comparison | Statistical test | n | Statistic | p-value | Outliers? | Figure |
|---|---|---|---|---|---|---|---|
| A | CD68 across development: Males | one-way ANOVA | 4 animals; 2 sections/animal; 1-3 zstacks/section | $F_{(3,61)}=4.58$ | 0.006 | P20: 1; P30: 1; P54: 1 | Fig. 2b |
| | P20:P30 | Holm-Sidak's posthoc | 12:17 | $t_{(61)}=2.66$ | 0.049 | | |
| | P20:P38 | Holm-Sidak's posthoc | 12:21 | $t_{(61)}=2.28$ | 0.078 | | |
| | P20:P54 | Holm-Sidak's posthoc | 12:15 | $t_{(61)}=0.08$ | 0.936 | | |
| | P30:P38 | Holm-Sidak's posthoc | 17:21 | $t_{(61)}=0.54$ | 0.832 | | |
| | P30:P54 | Holm-Sidak's posthoc | 17:15 | $t_{(61)}=2.92$ | 0.029 | | |
| | P38:P54 | Holm-Sidak's posthoc | 21:15 | $t_{(61)}=2.54$ | 0.054 | | |
| B | CD68+ D1r content across development: Males | one-way ANOVA | 4 animals; 2 sections/animal; 1-3zstacks/section | $F_{(3,60)}=3.89$ | 0.013 | P20: 1; P30: 1; P38: 1; P54: 1 | Fig. 2b |
| | P20:P30 | Holm-Sidak's posthoc | 12:17 | $t_{(60)}=2.54$ | 0.066 | | |
| | P20:P38 | Holm-Sidak's posthoc | 12:20 | $t_{(60)}=1.93$ | 0.166 | | |
| | P20:P54 | Holm-Sidak's posthoc | 12:15 | $t_{(60)}=0.10$ | 0.921 | | |
| | P30:P38 | Holm-Sidak's posthoc | 17:20 | $t_{(60)}=0.77$ | 0.689 | | |
| | P30:P54 | Holm-Sidak's posthoc | 17:15 | $t_{(60)}=2.81$ | 0.039 | | |
| | P38:P54 | Holm-Sidak's posthoc | 20:15 | $t_{(60)}=2.17$ | 0.129 | | |
| C | CD68+ C3 content across development: Males | one-way ANOVA | 4 animals; 2 sections/animal; 1-3 zstacks/section | $F_{(3,61)}=1.35$ | 0.267 | P30: 1; P38: 1; P54: 1 | Fig. 2c |
| D | CD68+ C3-D1r content across development: Males | one-way ANOVA | 4 animals; 2 sections/animal; 1-3 zstacks/section | $F_{(3,60)}=5.70$ | 0.002 | P20: 1; P30: 1; P38: 1; P54: 1 | Fig. 2d |
| | P20:P30 | Holm-Sidak's posthoc | 12:17 | $t_{(60)}=2.79$ | 0.028 | | |
| | P20:P38 | Holm-Sidak's posthoc | 12:20 | $t_{(60)}=0.09$ | 0.933 | | |
| | P20:P54 | Holm-Sidak's posthoc | 12:15 | $t_{(60)}=0.77$ | 0.714 | | |
| | P30:P38 | Holm-Sidak's posthoc | 17:20 | $t_{(60)}=3.10$ | 0.015 | | |
| | P30:P54 | Holm-Sidak's posthoc | 17:15 | $t_{(60)}=3.81$ | 0.002 | | |
| | P38:P54 | Holm-Sidak's posthoc | 20:15 | $t_{(60)}=0.96$ | 0.714 | | |
| E | CD68 across development: Females | one-way ANOVA | 4 animals; 2 sections/animal; 1-3 zstacks/section | $F_{(3,41)}=7.59$ | <0.001 | P20: 1; P38: 1 | Fig. 2e |
| | P20:P30 | Holm-Sidak's posthoc | 8:12 | $t_{(41)}=3.25$ | 0.012 | | |
| | P20:P38 | Holm-Sidak's posthoc | 8:10 | $t_{(41)}=0.95$ | 0.346 | | |
| | P20:P54 | Holm-Sidak's posthoc | 8:15 | $t_{(41)}=1.35$ | 0.334 | | |
| | P30:P38 | Holm-Sidak's posthoc | 12:10 | $t_{(41)}=4.52$ | <0.001 | | |
| | P30:P54 | Holm-Sidak's posthoc | 12:15 | $t_{(41)}=2.30$ | 0.078 | | |
| | P38:P54 | Holm-Sidak's posthoc | 10:15 | $t_{(41)}=2.56$ | 0.056 | | |
| F | CD68+ D1r content across development: Females | one-way ANOVA | 4 animals; 2 sections/animal; 1-3 zstacks/section | $F_{(3,40)}=3.81$ | <0.001 | P20: 1; P38: 1 | Fig. 2f |
| | P20:P30 | Holm-Sidak's posthoc | 8:12 | $t_{(40)}=3.13$ | 0.016 | | |
| | P20:P38 | Holm-Sidak's posthoc | 8:10 | $t_{(40)}=0.89$ | 0.381 | | |
| | P20:P54 | Holm-Sidak's posthoc | 8:14 | $t_{(40)}=1.58$ | 0.228 | | |
| | P30:P38 | Holm-Sidak's posthoc | 12:10 | $t_{(40)}=4.32$ | <0.001 | | |
| | P30:P54 | Holm-Sidak's posthoc | 12:14 | $t_{(40)}=1.85$ | 0.201 | | |
| | P38:P54 | Holm-Sidak's posthoc | 10:14 | $t_{(40)}=2.71$ | 0.039 | | |
| G | CD68+ C3 content across development: Females | one-way ANOVA | 4 animals; 2 sections/animal; 1-3 zstacks/section | $F_{(3,32)}=9.39$ | <0.001 | P30: 1 | Fig. 2g |
| | P20:P30 | Holm-Sidak's posthoc | 6:8 | $t_{(32)}=4.42$ | <0.001 | | |
| | P20:P38 | Holm-Sidak's posthoc | 6:10 | $t_{(32)}=4.46$ | <0.001 | | |
| | P20:P54 | Holm-Sidak's posthoc | 6:12 | $t_{(32)}=2.19$ | 0.071 | | |
| | P30:P38 | Holm-Sidak's posthoc | 8:10 | $t_{(32)}=0.17$ | 0.864 | | |
| | P30:P54 | Holm-Sidak's posthoc | 8:12 | $t_{(32)}=2.83$ | 0.032 | | |
| | P38:P54 | Holm-Sidak's posthoc | 10:12 | $t_{(32)}=2.83$ | 0.032 | | |
| H | CD68+ C3-D1r content across development: Females | one-way ANOVA | 4 animals; 2 sections/animal; 1-3 zstacks/section | $F_{(3,38)}=0.11$ | 0.160 | P30: 1; P38: 1 | Fig. 2h |

Statistical details for every analysis in Fig. 2

Table 2G). Despite significant D1r and C3 expression in CD68+ lysosomes, there was no change in colocalized C3-D1r content (Fig. 2i; Table 2H). Together, these data support the notion that C3-tagging and microglial immune processes may be regulating D1r levels in males during adolescence, while in females there may be meaningful interactions between D1r, C3, and microglia, but the data are not yet clear if those processes are related to one another.

**C3–C3R interactions mediate sex-specific D1r elimination**. Neutrophil inhibitor factor (NIF) is a well-characterized canine hookworm peptide that binds specifically to the CD11b subunit of C3R, occluding its ability to bind its natural ligands[26,27]. NIF has been used systemically to improve stroke outcomes in a rat model[28], treat diabetic retinopathy in mice[29], and as an exploratory treatment for stroke in humans[30]. In vitro, NIF binds with high affinity to CD11b and is irreversible in some experimental conditions[27]. Indeed, in humans, a single infusion of 1.0–1.5 mg/kg recombinant NIF (UK-279,276) can saturate over 80% of CD11b receptors in serum for at least 7 days[30]. However, to our knowledge, whether NIF can affect microglia has not been assessed. We first tested whether NIF can inhibit phagocytosis in microglia ex vivo. Microglia were isolated from whole brain and treated with media containing either vehicle, 60 ng NIF (reconstituted according to manufacturer's recommendations at 200 μg/mL; 1×), or 120 ng NIF (reconstituted at 400 μg/mL; 2×). Fluorescent carboxylate microspheres (hereafter referred to as beads) were applied to induce phagocytic activity. After 90 mins, phagocytic activity was significantly impaired when microglia were exposed to 60 ng, but not 120 ng of NIF (Fig. 3A; Table 3A). In a separate set of ex vivo experiments without beads (Fig. 3B), there were no changes in NIF (Fig. 3C; Table 3B) or CD11b (Fig. 3D; Table 3C) immunoreactivity for up to 120 min in microglia exposed to 60 ng NIF, indicating that at least under

these conditions, NIF treatment did not result in NIF-CD11b complex endocytosis and degradation.

If D1r downregulation in the NAc results from C3 tagging and microglial phagocytosis, then blocking the ability of microglial C3R to recognize and bind C3 should disrupt the normal developmental elimination of D1rs. To examine this hypothesis, we utilized a within-animal design: we pharmacologically disrupted C3–C3R interactions in the NAc with NIF in one hemisphere, while the contralateral control NAc was treated with vehicle (sterile PBS), thus allowing it to undergo normal developmental D1r downregulation processes. NIF (60 ng in 300 nL) was microinjected into the dorsomedial NAc of one hemisphere, and vehicle into the contralateral hemisphere of either P30 males or P22 females (Supplementary Fig. 2A, B), ages at which D1r downregulation has not yet occurred in each sex. Tissue was collected 8 days later, at P38 and P30 in males and females, respectively (Fig. 3e, g). This age was chosen for assessment because it is an age at which each sex should have undergone developmental downregulation of D1rs; thus if there are changes in NAc D1r levels by virtue of impaired C3–C3R interactions, they should be apparent when compared with the contralateral control hemisphere. In males, D1r immunoreactivity in the NAc of NIF-treated hemispheres were on average ~25%

higher than contralateral vehicle-treated hemispheres (Fig. 3f; Table 3D), while there was no difference in D1r levels between NIF and vehicle-treated hemispheres in females (Fig. 3h; Table 3E; Supplementary Fig. 2F,G). An important caveat should be noted: NIF was cleared in vivo 3–5 days after microinjection (Supplementary Fig. 2E), and thus how NIF was cleared and whether NIF clearance on its own caused changes in D1r regulation in the NAc is unclear. These data demonstrate for the first time a sex-specific immune mechanism regulating dopaminergic NAc development during adolescence.

**Sex-specific social behaviors require C3–C3R interactions.** We next sought to determine if D1r downregulation, and thus the mechanisms which regulate this process, could have consequences for normal developmental changes in adolescent behavior. Social behavior, and in particular, social play behavior is known to peak during adolescence and requires dopamine signaling in the NAc[7]. Moreover, recent dissection of the neural circuitry supporting pro-social behavior has implicated D1rs in the NAc as a key substrate[8,9]. To first assess if social behaviors change over the adolescent ages in which we characterized molecular development in the NAc, separate cohorts of experimental male and female rats were single-housed for 24 h, a period

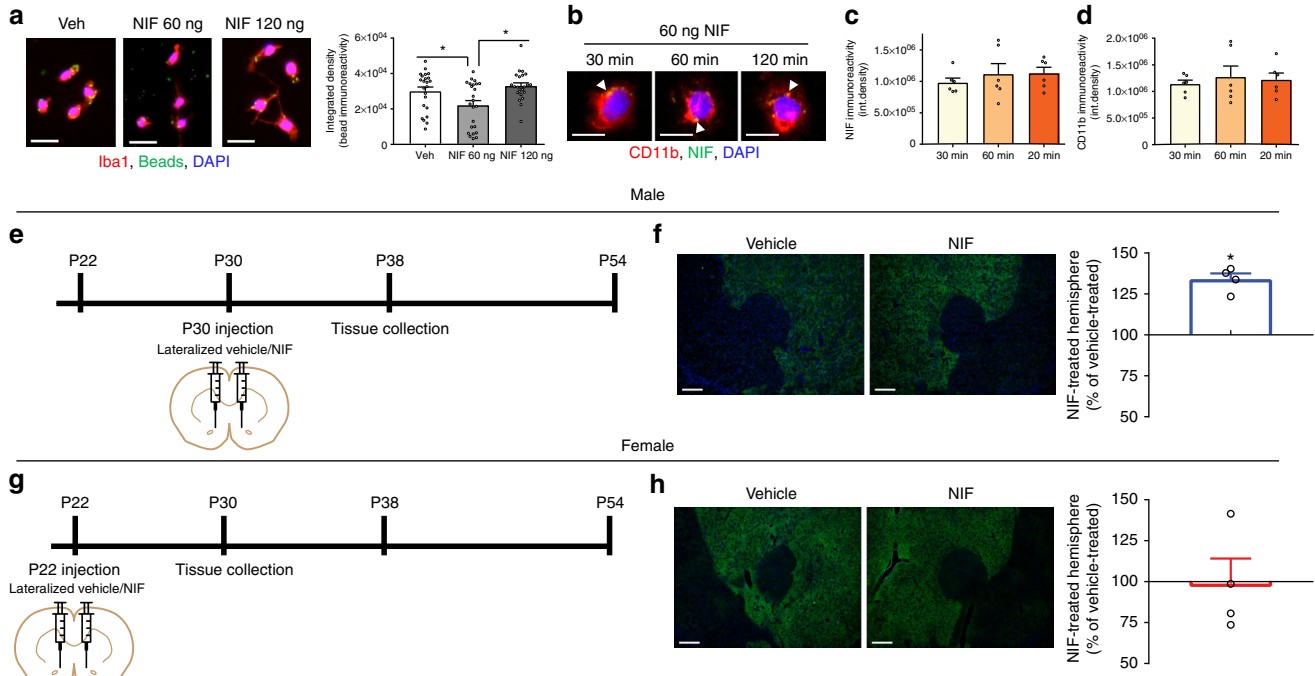

**Fig. 3** C3–C3R interactions mediate developmentally-typical D1r elimination in vivo in males, but not females. Neutrophil inhibitor factor (NIF), a peptide that binds specifically to the CD11b subunit of C3 receptors (C3Rs), was assessed for its efficacy in reducing microglial phagocytic activity ex vivo. **a** In microglia isolated from whole brain, 60 ng, but not 120 ng NIF inhibited microglial phagocytosis of fluorescent beads (Table 3A). Representative images precede histograms; scale bar equals 20 μm. $n = 4$, with 2 replications/condition. **b** Microglia were isolated from whole brain and incubated with 60 ng NIF, and then immunocytochemically assessed for NIF and CD11b at 30, 60, and 120 min. Closed arrow heads indicate NIF (green) immunoreactivity. Scale bar equals 10 μm. There was no change in **c** NIF or (Table 3B) **d** CD11b (Table 3C) immunoreactivity over 2 h, suggesting that NIF was not causing the degradation of its receptor. $n = 3$; 2 replications/condition. To determine if developmental D1r downregulation requires C3–C3R interactions, NIF or Vehicle was microinjected in the NAc at **e** P30 in males and **g** P22 in females (both represent sex-specific ages prior to D1r downregulation), and then tissue was assessed at P38 and P30, respectively, an age at which D1rs should be developmentally downregulated. **f** In males, NIF-treated hemispheres exhibited significantly more (~25%) D1r immunoreactivity than within-animal vehicle-treated hemispheres (Table 3D). **h** In females, NIF-treated hemispheres exhibited the same D1r immunoreactivity as within-animal vehicle-treated hemispheres (Table 3E). Representative images precede histograms; scale bars equal 100 μm. $n = 4$/sex with counterbalanced injections. For ex vivo experiments, data were analyzed with one-way ANOVAs and Holm-Sidak's post-hoc comparisons. For in vivo experiments, D1r data from 4–7 different sections per animal were averaged and then calculated as a percentage of within-animal vehicle control levels. Data were analyzed with 2-tailed one-sample $t$-tests. Histograms portray the mean ± SEM with individual data points overlaid. Significant post-hoc Holm-Sidak $t$-tests (**a**–**d**) and one-sample $t$-tests from 100 (**f, h**) ($p < 0.05$) comparisons are delineated with an asterisk. All statistics are in Table 3

**Table 3 Detailed statistics corresponding with Fig. 3**

| Comparison | Statistical test | n | Statistic | p-value | Outliers? | Figure |
|---|---|---|---|---|---|---|
| *A* | | | | | | |
| Ex vivo microglial bead phagocytosis | One-way ANOVA | Four animals; two replications/condition; 2–3 z-stacks/replication | $F_{(2,67)} = 6.24$ | 0.003 | N/A | Fig. 3a |
| Vehicle:NIF 60 ng | Holm-Sidak's post-hoc | 23:24 | $t_{(67)} = 2.49$ | 0.030 | | |
| Vehicle:NIF 120 ng | Holm-Sidak's post-hoc | 23:23 | $t_{(67)} = 0.91$ | 0.370 | | |
| NIF 60 ng:NIF 120 ng | Holm-Sidak's post-hoc | 24:23 | $t_{(67)} = 3.41$ | 0.003 | | |
| *B* | | | | | | |
| Ex vivo NIF timecourse: NIF immunoreactivity | One-way ANOVA | Three animals; two replications/condition; three z-stacks/replication | $F_{(2,15)} = 0.63$ | 0.295 | N/A | Fig. 3c |
| *C* | | | | | | |
| Ex vivo NIF timecourse: CD11b immunoreactivity | One-way ANOVA | Three animals; two replications/condition; three z-stacks/replication | $F_{(2,15)} = 0.21$ | 0.265 | N/A | Fig. 3d |
| *D* | | | | | | |
| In vivo matched NIF-Vehicle injection: male | One sample *t*-test (from 100) | Four animals; 4–7 sections/animal | $t_{(3)} = 9.24$ | 0.003 | N/A | Fig. 3F |
| *E* | | | | | | |
| In vivo matched NIF-Vehicle injection: female | One sample *t*-test (from 100) | Four animals; 5–6 sections/animal | $t_{(3)} = 0.07$ | 0.947 | N/A | Fig. 3H |

Statistical details for every analysis in Fig. 3

of isolation known to increase social interaction and play without inducing long-lasting detrimental effects[31,32]. The following day, a novel age-matched and sex-matched rat was introduced into the home cage of the experimental rat, and their interactions were recorded for 10 min (a "resident intruder" paradigm;[33,34] Fig. 4a). Social play and social exploration (i.e., non-play social behavior consisting of allogrooming and sniffing of the conspecific) initiated by the experimental animal was scored by an experimenter blind to age and sex conditions. Strikingly, male social play behavior mirrored the pattern of NAc D1r expression: social play transiently peaked at P30 (Fig. 4b; Table 4A), with no changes in social exploration (Fig. 4c; Table 4B). Increased total play behavior was accompanied by increased incidence of classic play postures in the experimental animal, including nape attacks and supine postures (Supplementary Fig. 3A-E). In contrast, play levels did not significantly change over time in females (Fig. 4D; Table 4C; Supplementary Fig. 3F-J). Rather, female social exploration increased at P30 and then decreased again at P54 (Fig. 4e; Table 4D). Neither weight differences between the experimental and control animals nor estrous cycle where applicable in females (P38 and 54) correlated with behavior (Supplementary Fig. 3K-M).

These data demonstrate a behavioral correlate, specifically social play behavior, to NAc D1r expression patterns in males during adolescent development. While behavior did change over adolescence in females and warrants further investigation, there was no overt correlation with the pattern of D1r expression in the NAc. To determine if there was a causal link between the male-specific immune-mediated D1r elimination we observed in the NAc and male-specific changes in social play during adolescence, we bilaterally microinjected NIF or vehicle into the NAc of males at P30 and females at P22, and then assessed social behavior at P38 and P30, respectively (Fig. 4f). Male social play behavior (Fig. 4g; Table 4E), but not social exploration (Fig. 4h; Table 4F), was significantly increased in NIF-treated relative to vehicle-treated males. While there was no significant change in discrete play postures when analyzed separately, the change in play behavior was primarily characterized by increases in the number of nape attacks and supine postures exhibited by NIF-treated animals (Supplementary Fig. 4A-E). To our surprise, NIF-treated females also demonstrated modestly (but significantly) increased social play (Fig. 4i; Table 4G), but not social exploration behavior

(Fig. 4j; Table 4H). NIF-treated females also exhibited more nape attacks and miscellaneous play elicitations (Supplementary Fig. 4f–j). There was a significant correlation between weight and social play in vehicle-treated females, but no other correlations between weight and social behaviors (Supplemental Fig. 4K–N). NIF microinjections into the NAc at P54 (Supplementary Fig. 2C, D) had no effect on social behaviors in either males or females (Supplementary Fig. 5), indicating that the behavioral efficacy of NIF is contingent on C3–C3R interactions during active periods of development.

**C3–C3R regulation of D1rs impacts male social play**. In a final set of experiments, we determined if the behavioral consequences of NAc NIF treatment on social play behavior were D1r dependent. NIF was co-injected with siRNA against rat D1r (200 μM; NIF si) in one hemisphere, and control scrambled RNA (scRNA) in the contralateral hemisphere (NIF sc) at P30 in males and P22 in females (Fig. 5a, c). Despite the presence of NIF, D1r levels were decreased in the siRNA treated hemispheres of all subjects when D1r immunoreactivity was assessed 8 days later in both sexes (P38 in males and P30 in females; Fig. 5b, d; Table 5A, B). Next, we bilaterally co-injected NIF/vehicle with sc/siRNA against D1r into the NAc in P30 males or P22 females, and then tested social behavior as described at P38 and P30, respectively (Fig. 5E). Replicating the previous findings, using NIF to inhibit C3–C3R interactions at P30 (Fig. 5e) increased social play behavior at P38 in males, but this effect was blocked by siRNA against rat D1r (Fig. 5f; Table 5C; Supplementary Fig. 6A-E), with no changes in social exploration in any group (Fig. 5g; Table 5D). In females, the unexpected increase in social play at P30 by P22 NIF injections was not prevented by siRNA against rat D1r, indicating this effect on play is independent of D1r (Fig. 5h; Table 5E; Supplementary Fig. 6F-J), again with no changes in social exploration (Fig. 5i; Table 5F). Collectively, this report demonstrates for the first time a developmentally-restricted or age-restricted and sex-specific role for immune signaling and microglia in developmental changes in behavior, with C3–C3R regulation of D1rs in the NAc mediating overt developmental changes in social play behavior in males, and C3–C3R regulation of an as yet undetermined process impacting basal levels of social play behavior in females.

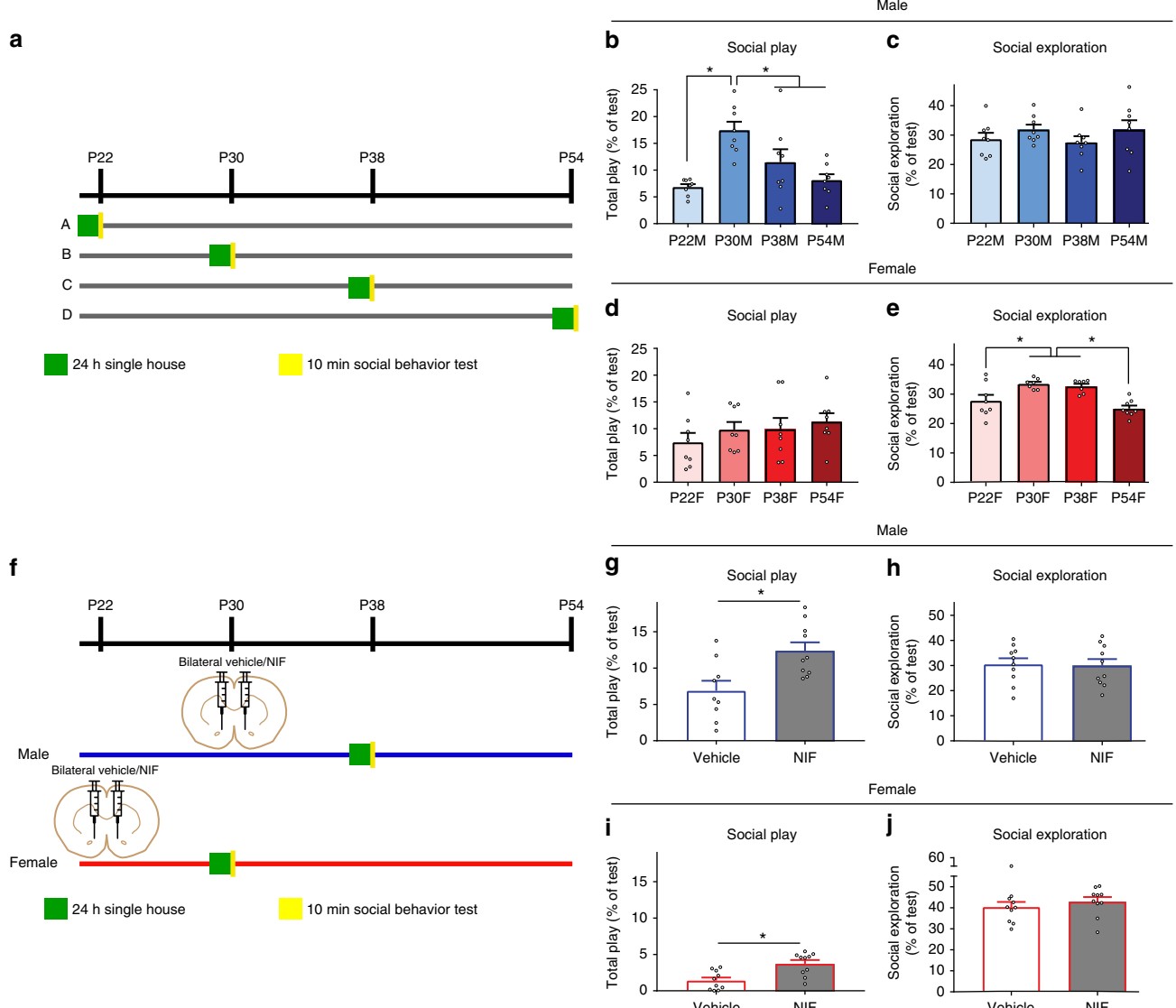

**Fig. 4** Sex-specific social behavior patterns over adolescence require C3–C3R interactions. **a** Experimental design: separate cohorts of experimental animals ($n = 8$/sex/age) were single-housed for 24 h prior to test, after which time a novel age-matched and sex-matched conspecific was introduced into their home cage. Ten minutes of interactions were recorded, and later an experimenter blinded to the conditions coded the interactions for either total play or social exploration (i.e., non-play social behavior). Only behaviors initiated by the experimental animal were considered. Males: **b** Male social play behavior transiently increased at P30 (Table 4A), **c** with no changes in social exploration over development (Table 4B). Females: **d** Female social play behavior did not change over development (Table 4C), **e** while social exploration peaked from P30 to 38 (Table 4D). **f** To determine if C3–C3R interactions regulating D1r levels between P30 and 38 in males is required for the decline in social play behavior observed at this time, NIF or vehicle was injected bilaterally in P30 males or P22 females ($n = 10$/sex/treatment). Animals were then single housed for 24 h at P37 or P29, and assessed for social behavior as described at P38 and P30, respectively. Males: **g** Interrupting C3–C3R interactions at P30 increased social play behavior at P38 (Table 4E), **h** without affecting social exploration (Table 4F). Females: **i** Female social play behavior, while at a low basal level, also modestly increased at P30 after P22 NIF injection (Table 4G), **j** with no changes in social exploration (Table 4H). Data were analyzed either with one-way ANOVAs and Holm-Sidak's post-hoc comparisons or with two-tailed unpaired *t*-tests. Histograms portray the mean ± SEM with individual data points overlaid. Significant post-hoc Holm-Sidak *t*-tests (**b–e**) and unpaired *t*-tests (**g–j**) ($p < 0.05$) comparisons are delineated with an asterisk. All statistics are in Table 4

## Discussion

Herein, we describe a role for microglia and immune signaling in sex-specific dopaminergic development in the NAc and social behavior during adolescence. D1rs in the NAc are downregulated during adolescent development in males, between early-mid adolescence (P30–38), and in females between the peri-early adolescent period (P20–30). While both sexes undergo D1r downregulation, the mechanisms by which this is accomplished are starkly different: in males, interactions between complement

C3 and the microglia-specific C3R mediate the engulfment and lysosomal elimination of D1rs. However, D1r downregulation in females appears to be independent of complement C3 and microglia. D1r expression patterns in the NAc mirror social play behavior in males (but not females), raising the possibility that D1r expression and the immune mechanisms which regulate it could be causally linked with developmental changes in behavior. Indeed, pharmacological interference of C3–C3R interactions decreases microglial phagocytic activity ex vivo, increases D1r

**Table 4 Detailed statistics corresponding with Fig. 4**

| Comparison | Statistical test | n | Statistic | p-value | Outliers? | Figure |
|---|---|---|---|---|---|---|
| *A* | | | | | | |
| Social play across development: males | One-way ANOVA | Eight animals/age | $F_{(3,28)} = 9.03$ | <0.001 | N/A | Fig. 4b |
| P22:P30 | Holm-Sidak's post-hoc | 8:8 | $t_{(28)} = 4.76$ | <0.001 | | |
| P22:P38 | Holm-Sidak's post-hoc | 8:8 | $t_{(28)} = 2.10$ | 0.130 | | |
| P22:P54 | Holm-Sidak's post-hoc | 8:8 | $t_{(28)} = 0.59$ | 0.559 | | |
| P30:P38 | Holm-Sidak's post-hoc | 8:8 | $t_{(28)} = 2.67$ | 0.049 | | |
| P30:P54 | Holm-Sidak's post-hoc | 8:8 | $t_{(28)} = 4.17$ | 0.001 | | |
| P38:P54 | Holm-Sidak's post-hoc | 8:8 | $t_{(28)} = 1.50$ | 0.267 | | |
| *B* | | | | | | |
| Social exploration across development: males | One-way ANOVA | Eight animals/age | $F_{(3,28)} = 0.90$ | 0.456 | N/A | Fig. 4c |
| *C* | | | | | | |
| Social play across development: females | One-way ANOVA | Eight animals/age | $F_{(3,28)} = 0.84$ | 0.484 | N/A | Fig. 4d |
| *D* | | | | | | |
| Social exploration across development: females | One-way ANOVA | Eight animals/age | $F_{(3,26)} = 9.30$ | <0.001 | P30: 1; P38: 1 | Fig. 4e |
| P22:P30 | Holm-Sidak's post-hoc | 8:7 | $t_{(26)} = 3.10$ | 0.019 | | |
| P22:P38 | Holm-Sidak's post-hoc | 8:7 | $t_{(26)} = 2.64$ | 0.041 | | |
| P22:P54 | Holm-Sidak's post-hoc | 8:8 | $t_{(26)} = 1.47$ | 0.286 | | |
| P30:P38 | Holm-Sidak's post-hoc | 7:7 | $t_{(26)} = 0.46$ | 0.659 | | |
| P30:P54 | Holm-Sidak's post-hoc | 7:8 | $t_{(26)} = 4.51$ | 0.001 | | |
| P38:P54 | Holm-Sidak's post-hoc | 7:8 | $t_{(26)} = 4.05$ | 0.002 | | |
| *E* | | | | | | |
| Social play: vehicle vs. NIF treated males | Unpaired *t*-test | Ten animals/treatment | $t_{(17)} = 3.12$ | 0.006 | Veh: 1 | Fig. 4g |
| *F* | | | | | | |
| Social exploration: vehicle vs. NIF treated males | Unpaired *t*-test | Ten animals/treatment | $t_{(18)} = 0.88$ | 0.880 | N/A | Fig. 4h |
| *G* | | | | | | |
| Social play: vehicle vs. NIF treated females | Unpaired *t*-test | Ten animals/treatment | $t_{(18)} = 3.60$ | 0.002 | N/A | Fig. 4i |
| *H* | | | | | | |
| Social exploration: vehicle vs. NIF treated females | Unpaired *t*-test | Ten animals/treatment | $t_{(18)} = 0.43$ | 0.807 | N/A | Fig. 4j |

Statistical details for every analysis in Fig. 4

levels in vivo, and increases social play in a D1r-dependent manner in males. Surprisingly, though C3–C3R interactions are not regulating D1r levels in females and levels of play are overall lower compared to males, perturbing C3R with NIF also increases social play behavior in females, but this occurs in a D1r-independent manner.

The dopaminergic mesolimbic reward circuitry plays a central role in a broad repertoire of cognitive, social, and motor functions, thus linking abnormalities in this system to a wide variety of neurodevelopmental, neuropsychiatric, and neurodegenerative diseases. Intriguingly, a number of neuropsychiatric disorders commonly emerge in adolescence, including depression, anxiety, schizophrenia, and addictive behaviors, many of which present with a striking sex bias[35,36]. This convergence suggests that there may be instances in which adolescent development is misdirected from a normal, healthy state to a disordered, pathologic state, potentially in sex-specific ways. However, the molecular mechanisms underlying reward circuitry development in adolescence are not well known, and, consequently, what causes "misdirected development" or how it is mediated at the molecular level remains unclear. In particular, the neural circuitry underlying social behavior has garnered increasing attention with the recognition of social deficits as a common symptom in a variety of neurodevelopmental (i.e., autism) and neuropsychiatric (i.e., anxiety or schizophrenia) disorders. Consistent with our data, social play behavior has been reported to be most robust between P30–40 in rats, and generally (though not exclusively) rough-and-tumble play is higher in male than in female adolescents[37,38]. Interestingly, we did not see changes in female social play behavior over adolescence, but rather in social exploration, which we defined as allogrooming (social grooming) and conspecific sniffing. Social interactions in adolescence, and play behavior in

particular, are thought to be a way for males to 'practice' behaviors that result in hierarchical and sexual competency later in life[39,40]. In females, however, adolescent social interactions may serve roles that we are not adequately assessing. For instance, maternal behaviors and pup interaction are of critical importance to successful adult female behavior in rats, and increased pup-centered behaviors often coincides with adolescence in females[41]. Moreover, both the familiarity of a conspecific animal (i.e., a prior cage mate or novel female) as well as the social context (i.e., home cage or novel cage) have been demonstrated to influence female play behavior[33,42]. Thus, we propose that to understand the molecular mechanisms governing female NAc (and greater reward circuitry) development, we first need to identify the repertoire of experiences that are most salient to that sex. The importance of changes in social exploration observed here will provide an excellent foundation on which to begin this exploration.

Many sex differences in neurodevelopment and behavior, including sex-specific social behavior, are established by a surge of male sex hormones (androgens) in male, but not female fetuses around the time of birth[37,43]. Remarkably, in regions of the brain known to develop highly sexually dimorphic neural architecture, the effects of androgen masculinization at birth is mediated indirectly through microglia and immune signaling[44,45]. Consistent with the engagement of microglia and immune signaling specifically in the male brain early in life to mediate sexual organization, a recent report indicates that the human transcriptome in neocortex displays higher phagocytosis-related and immune-related gene expression in males than in females prior to birth, which then switches to a higher female bias after birth[46]. Intriguingly, in this same dataset, there was another age at which males again demonstrated higher immune gene expression

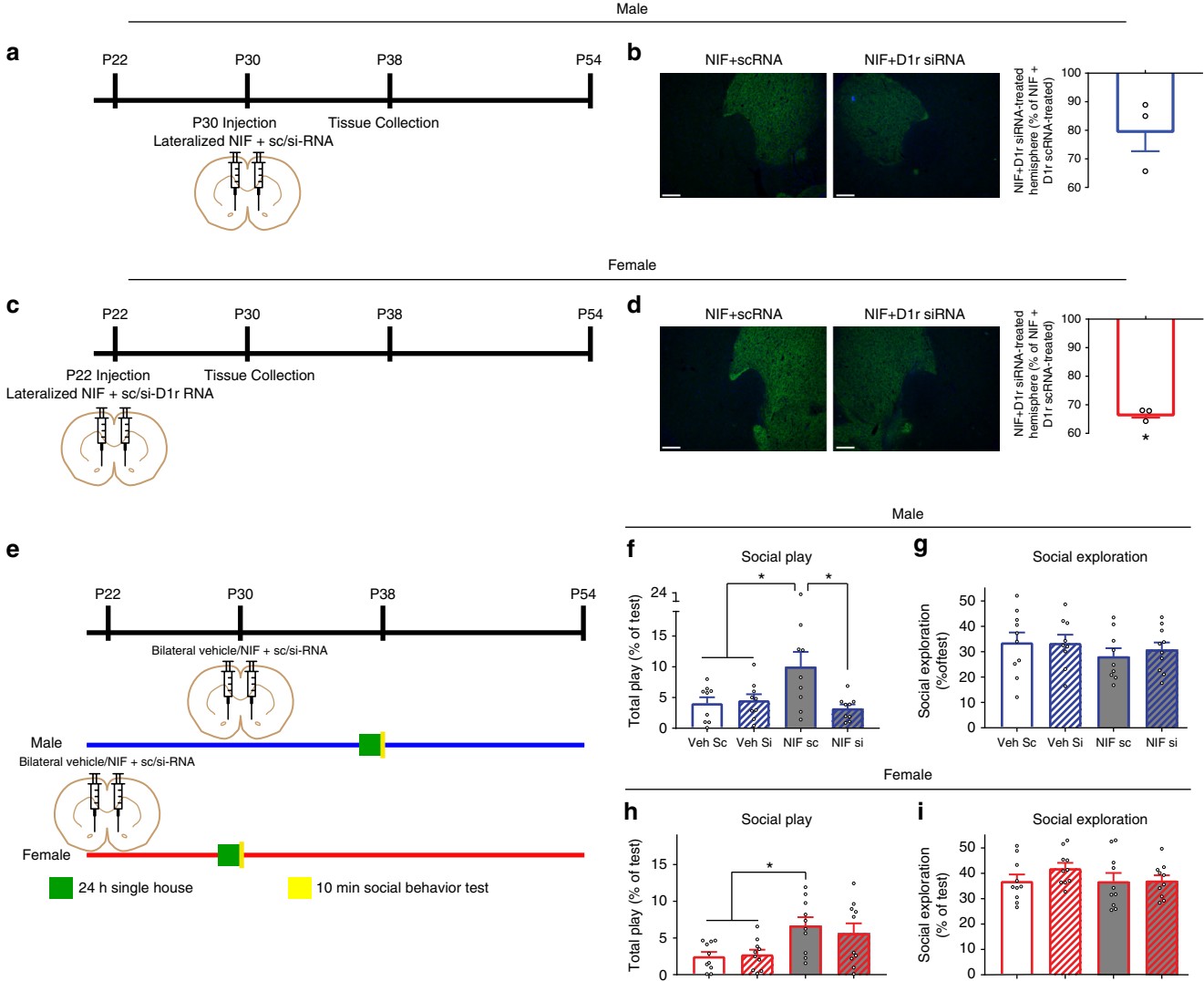

**Fig. 5** C3–C3R regulation of D1rs regulates male, but not female, social play behavior. To determine if disrupting C3–C3R interactions via NIF increases play in a D1r-dependent manner, we utilized siRNA against rat D1r (200 μM) or scRNA controls. **a** P30 males were co-injected with NIF + D1r siRNA in one hemisphere and NIF + scRNA in the contralateral hemisphere, and then D1r immunoreactivity was assessed at P38. **b** In the presence of NIF, D1r siRNA was capable of decreasing D1r levels relative to NIF + scRNA within-animal control hemispheres (~20%; Table 5A). **c** P22 females were injected with NIF + D1r siRNA in one hemisphere and NIF + scRNA in the contralateral hemisphere, and then D1r immunoreactivity was assessed at P30. **d** In the presence of NIF, D1r siRNA was capable of decreasing D1r levels relative to NIF + scRNA within-animal control hemispheres (~30%; Table 5B). Representative images precede histograms; scale bar equals 100 μm. $n = 3$/sex. **e** Experimental design: P30 males and P22 females were bilaterally microinjected into the NAc with either Vehicle + scRNA (Veh sc), Vehicle + D1r siRNA (Veh si), NIF + scRNA (NIF sc), or NIF + D1r siRNA (NIF si). At P37 and P22, animals were single-housed for 24 h, and then a novel age- and sex-matched conspecific was introduced into their home cage for 10 min social behavior tests ($n = 9–10$/sex/treatment). Males: **f** NIF + scRNA increased social play behavior in males, which was eliminated by D1r siRNA (NIF si; Table 5C). **g** No treatment changed social exploration behavior (Table 5D). Females: **h** NIF + scRNA increased social play behavior in females, which was not affected by D1r siRNA (NIF si; Table 5E). **i** No treatment changed social exploration behavior (Table 5F). Immunohistochemical data were calculated as in Fig. 2 and analyzed with two-tailed one-sample $t$-tests. Behavioral data were analyzed with one-way ANOVAs and Holm-Sidak's post-hoc comparisons. Histograms portray the mean ± SEM with individual data points overlaid. Significant one-sample $t$-tests from 100 (**b**, **d**) and post-hoc Holm-Sidak $t$-tests (**f–i**) ($p < 0.05$) comparisons are delineated with an asterisk. All statistics are in Table 5

relative to females: late childhood to early-adolescence. Here, we demonstrate that the male, but not female NAc, engages classical immune signaling to refine dopaminergic circuitry during adolescent development. Collectively, these data raise the possibility that the preferential engagement of immune-mediated mechanisms is pre-programmed, potentially via early-life masculinization events, to serve male development. Our data demonstrate that interfering with C3–C3R immune interactions with NIF blocks

developmentally-programmed D1r downregulation and increases social play behavior in a D1r-dependent manner in P30–38 adolescent males. While D1r downregulation in females did not require C3–C3R interactions, and females did not demonstrate any overt changes in social play behavior over the course of adolescence, P22 NIF treatment nevertheless increased social play behavior, in a D1r-independent manner. Because behavioral changes only arise in females when the molecular milieu of the

**Table 5 Detailed statistics corresponding with Fig. 5**

| Comparison | Statistical test | n | Statistic | p-value | Outliers? | Figure |
|---|---|---|---|---|---|---|
| A | | | | | | |
| In vivo matched NIF + D1r siRNA-NIF + scRNA injection: male | One sample t-test (from 100) | Three animals; eight sections/animal | $t_{(2)} = 2.81$ | 0.107 | N/A | Fig. 5b |
| B | | | | | | |
| In vivo matched NIF + D1r siRNA-NIF + scRNA injection: female | One sample t-test (from 100) | Three animals; eight sections/animal | $t_{(2)} = 27.1$ | 0.001 | N/A | Fig. 5d |
| C | | | | | | |
| Social play: NIF/Vehicle + si/scRNA males | One-way ANOVA | 9–10 animals/treatment | $F_{(3,32)} = 3.84$ | 0.019 | Veh sc: 1; NIF si: 1 | Fig. 5f |
| Veh sc: Veh si | Holm-Sidak's post-hoc | 9:9 | $t_{(32)} = 0.24$ | 0.902 | | |
| Veh sc: NIF sc | Holm-Sidak's post-hoc | 9:9 | $t_{(32)} = 2.95$ | 0.029 | | |
| Veh sc: NIF si | Holm-Sidak's post-hoc | 9:9 | $t_{(32)} = 0.41$ | 0.902 | | |
| Veh si: NIF sc | Holm-Sidak's post-hoc | 9:9 | $t_{(32)} = 2.71$ | 0.042 | | |
| Veh si: NIF si | Holm-Sidak's post-hoc | 9:9 | $t_{(32)} = 0.65$ | 0.891 | | |
| NIF sc: NIF si | Holm-Sidak's post-hoc | 9:9 | $t_{(32)} = 3.35$ | 0.012 | | |
| D | | | | | | |
| Social exploration: NIF/Vehicle + si/scRNA males | One-way ANOVA | 9–10 animals/treatment | $F_{(3,34)} = 0.56$ | 0.664 | N/A | Fig. 5g |
| E | | | | | | |
| Social play: NIF/Vehicle + si/scRNA females | One-way ANOVA | Ten animals/treatment | $F_{(3,36)} = 4.42$ | 0.010 | N/A | Fig. 5f |
| Veh sc: Veh si | Holm-Sidak's post-hoc | 10:10 | $t_{(36)} = 0.19$ | 0.845 | | |
| Veh sc: NIF sc | Holm-Sidak's post-hoc | 10:10 | $t_{(36)} = 2.98$ | 0.030 | | |
| Veh sc: NIF si | Holm-Sidak's post-hoc | 10:10 | $t_{(36)} = 2.23$ | 0.122 | | |
| Veh si: NIF sc | Holm-Sidak's post-hoc | 10:10 | $t_{(36)} = 2.80$ | 0.040 | | |
| Veh si: NIF si | Holm-Sidak's post-hoc | 10:10 | $t_{(36)} = 2.05$ | 0.137 | | |
| NIF sc: NIF si | Holm-Sidak's post-hoc | 10:10 | $t_{(36)} = 0.75$ | 0.706 | | |
| F | | | | | | |
| Social exploration: NIF/Vehicle + si/scRNA females | One-way ANOVA | Ten animals/treatment | $F_{(3,36)} = 0.95$ | 0.428 | N/A | Fig. 5i |

Statistical details for every analysis in Fig. 5

NAc is tampered with, in our case via local NIF injections, we hypothesize that these data demonstrate a homeostatic, maintenance role for C3–C3R interactions in the female NAc which modify basal levels of social play behavior during adolescent development. In contrast, males exhibit concurrent peaks in C3 and D1r, colocalized C3-D1r immunoreactivity contacted by microglia and internal to microglial lysosomes in the NAc, and a decline in social play behavior, all between P30–38. Moreover, we present evidence that C3–C3R interactions mediate the developmentally programmed decreases in both D1rs and social play behavior during adolescence in males. Thus female neural development may accomplish largely the same phenotypic and functional outcomes (i.e., appropriate D1r and social behavior maturation), by employing starkly different mechanisms, a phenomena also observed in sex differences in pain processing[47] and analgesia[48]. As with the interpretation of our behavioral data discussed above, we suggest that to understand sex-specific development at a molecular and behavioral level, we may need sex-specific experimental approaches and techniques.

Importantly, other neuromodulator/neurotransmitter systems are regulated in adolescence[49], and neuropeptides commonly associated with social behavior, like oxytocin, endocannabinoids, and opioids, have been demonstrated to modulate social behavior when manipulated locally in the amygdala, NAc, and VTA in rodents[8,50,51], as well as intra-nasally in humans[52]. Moreover, D1 is not the only important dopamine receptor mediating reward circuitry dynamics. In particular, D1rs and D2rs, are largely thought to be complementary. While both are G-protein coupled receptors that respond to dopamine, D1rs stimulate cAMP signaling, and thus have an "excitatory" effect in the cell; conversely, D2rs inhibit cAMP production, resulting in an "inhibitory"

effect[53]. Thus the ratio of these receptors is important for excitatory:inhibitory balance, and having a complete understanding of the sex-specific regulation of these complementary receptors will be important. Indeed, our data cannot rule out the possibility that there are changes in D1r transcription and translation, or compensatory changes in D2rs or other neural receptors, mediated by immune signaling. Additionally, there is a well-defined anatomical and functional dissociation between the shell and core of the NAc[54]. We defined our regions of interest not by core vs. shell, but by D1r immunoreactivity. Thus, while we are likely studying D1r dynamics in the core (i.e. immediately dorso-medial to the anterior commissure), we cannot exclude contributions from the shell. Interestingly, while core vs. shell distinctions are unquestionably important for some behaviors, dopamine signaling supporting social play does not appear to have a core vs. shell dependency[7]. Additionally, we focused our analyses primarily to the anterior NAc. There is an interesting anterior-posterior NAc dichotomy, in which the anterior NAc processes reward and posterior NAc processes aversion in humans[55] and rodents[56]. Given its different functions and the notion that adolescents experience both reward augmentation and dampened aversion[57], it will be important to determine if the development of the NAc is not unitary, but rather utilizing different mechanisms and/or different timescales along the anterior-posterior axis.

Finally, complement signaling involves a complex cascade of protein activation, either via the classical, lectin, or alternative pathways, all ultimately converging on the enzymatic cleavage of complement C3 into the iC3b fragment, which then acts as an opsonin and is a ligand of C3R/CD11b[58,59]. While it is clear that C3 and C3R are critical for neurodevelopment as well as engaged in pathology, how C3 expression and its proteolytic processing is

regulated in the brain is unknown. Importantly, the complement cascade is not the only means of eliminating synapses and refining neural circuitry. Other immune protein families like the major histocompatibility component (MHC) family[60], other cell types (astrocytes in particular)[61], C3R-independent synaptic remodeling and pre-synaptic trogocytosis by microglia[19], and immune-independent mechanisms (e.g., retraction due to competition[62] or GABA$_A$ receptor-mediated elimination[63]) have all been implicated in synaptic elimination. Interestingly, there is also evidence for neuro-glial communication guiding immune-mediated neural circuit refinement[64], and thus the molecular network which regulates C3-mediated elimination is likely to be far more complex than we currently appreciate. Irrespective of its upstream mediators, immune-mediated synaptic remodeling and elimination appears to be engaged in several models of neurological disorders including virus-induced cognitive dysfunction[24], healthy aging[23], and Alzheimer's disease[21,22], suggesting this natural developmental mechanism may be over-activated in pathology.

Our data raise many important questions, and point to several unexplored lines of inquiry for future exploration. Intriguingly, a recent report demonstrated different microglial signatures within different regions of the reward circuitry[65], and understanding how these structures develop in concert and cooperate to mediate behavioral outcomes will be revolutionary for our understanding of adolescent vulnerabilities to neuropsychiatric disorders. While challenging in practice, moving toward an experimental framework that assesses more than one signaling pathway and/or brain regions in parallel can reveal unappreciated cooperation in the service of behavior[66], and will ultimately be required to fully understand the development of this system.

## Methods

**Animals.** Either adult male and female Sprague-Dawley rats (both age postnatal day 70–75 (P70–75)) were purchased to be breeding pairs, or pregnant females were purchased ~7 days prior to giving birth (Harlan/Envigo, Dublin, VA). Animals were group-housed in individually ventilated cages with ad libitum access to food (Purina lab Diet 5001) and water. The colony was maintained at 23 °C on a 12:12 light-dark cycle (lights on at 07:00 h) and cages were changed once per week. Females were separated from males prior to birth. Litters were culled to a maximum of 12 pups per litter on P2–5, and at P21 pups were weaned into same sex pair-housing (2–4 rats per cage). For behavioral experiments, conspecific animals were purchased to be age- and sex-matched (Harlan/Envigo, Dublin, VA). PFA-fixed D1-tdTomato mouse brains were a generous gift from the Lobo lab at University of Maryland School of Medicine, and PFA-fixed D1r knock out and wildtype mouse brains were a generous gift from the Caron lab at Duke University. All experiments and animal care was approved by the Institutional Animal Care and Use Committees at Duke University and the Massachusetts General Hospital.

**Tissue collection.** Animals were euthanized at the ages indicated in each experiment by $CO_2$ anesthesia and exsanguination. Blood was collected prior to exsanguination via cardiac puncture, and then 0.9% saline was perfused until blood was cleared. Animals were decapitated and brains extracted. For immunohistochemical experiments, brains were post-fixed in 4% paraformaldehyde (PFA) in PBS. In surgically manipulated animals with immunohistochemical endpoints, perfusion with 4% PFA followed saline perfusion. Whole brains were then incubated in 4% PFA for 24 h at 4 °C. The remaining surgically manipulated animals were subjected to social play testing, and then euthanized after behavioral testing.

**Immunohistochemistry.** PFA-fixed brains were cryoprotected in 30% sucrose in 0.1 M PB for at least 2 days. Cryoprotected brains were rapidly frozen in molds with Tissue Tek on dry ice, and then cryosectioned at 20 μm. Four animals from different litters were represented in each experimental group. Sections were stored in 0.1 M PB with sodium azide (0.1%) until processing. After washing in PBS, sections were incubated at 80 °C for 30 min in 10 mM citric acid (pH 9.0) for epitope retrieval. Background fluorescence was quenched sequentially with 30 min incubations in 1 mg/mL sodium tetraborate in 0.1 M PB and then 50% methanol. Tissue was permeabilized for 1 h with 0.3% Triton-x100 in PBS and then blocked in 10% normal donkey serum (Jackson ImmunoResearch; West Grove, PA) in PBS for 1 h. Primary antibodies were applied sequentially for overnight incubations in 5% normal donkey serum in PBS at 4 °C: either anti-goat C3 (MP Biomedicals #0855713; Santa Ana, CA; 1:200), anti-mouse D1r (Novus Biologicals

#NB110–60017; Littleton, CO; 1:750), anti-rabbit Iba1 (Wako Chemicals #019–19741; Richmond, VA; 1:1500) or anti-rabbit CD68 (Abcam #ab125212; Cambridge, MA; 1:5000), anti-goat C3, anti-mouse D1r. Secondary antibodies were applied simultaneously for the C3-D1r-Iba1 triple stain (Donkey anti-goat Alexa Fluor (AF) 647, anti-mouse AF 488, anti-rabbit AF 563; Thermo Fisher Scientific, each 1:500) for 2 h at room temperature in the dark. For the CD68-C3-D1r triple stain, CD68 was amplified prior to proceeding with the subsequent primary incubations. Tissue was incubated with biotinylated anti-rabbit (Vector Laboratories; Burlingame, CA; 1:1000) for 2 h at room temperature, in Avidin-Biotin Complex solution for 30 min at room temperature (Vector Laboratories; Burlingame, CA), and then Tyramide Signal Amplification fluorescein reagent (Perkin-Elmer) for 10 mins at room temperature in the dark. After washing, the tissue was then incubated sequentially with C3 and D1r antibodies, and then incubated with secondary antibodies (anti-goat AF 563, anti-mouse AF 647; Thermo Fisher Scientific, 1:500) for 2 h at room temperature in the dark. All tissue was mounted on gelatin subbed slides, coverslipped with ProLong Antifade Mounting media (Thermo Fisher Scientific), sealed with nailpolish, and stored at −20 °C until imaging.

**Image acquisition and volumetric reconstructions.** One to three z-stacks (depending on the size of the ROI, see Supplementary Fig. 1B) per section within the D1r-rich region of the NAc were acquired on a Nikon A1SiR confocal microscope with 60x magnification and 0.1 μm step size ($n = 4$ animals per group, two sections per animal). Importantly, male and female tissue required different imaging parameters and thus are analyzed and graphically depicted separately. For these reasons, no direct comparisons between males and females can be made from these data. While exhibiting the same visual features used to select the NAc (i.e., rhinal fissure size and corpus callosum shape), female ROIs tended to be smaller than males, and thus fewer z-stacks were taken from female tissue (specific differences detailed in Table 1). Background was subtracted and threshold values were recorded using ImageJ (National Institutes of Health; Bethesda, MD). These files were then uploaded into Imaris (Bitplane; Zurich, Switzerland) to create volumetric reconstructions of (i) each individual channel, (ii) C3-D1r colocalization, and (iii) "masked" C3, D1r, and C3-D1r. The masking application refers to the use of one channel as a 3D outline, or mask, on which all other channels are superimposed. If other channels are within this 3D mask, they are considered in contact with or contained within the masking channel. For example, Iba1-masked D1r volume would constitute all D1r volumes that are either within (i.e., engulfed) or directly contacted by microglia, whereas a CD68-masked D1r volume constitutes all D1r volumes within a CD68+ lysosome. Total volumes for each channel reconstructed in Imaris (a maximum of 24 values across four different animals per group) were then used for statistical analysis.

**Ex vivo microglial assays.** For then phagocytosis assay, microglia were isolated from minced adult female brains ($n = 4$) according to Hanamsagar et al[67]. Briefly, tissue was enzymatically and mechanically disrupted and then myelin and debris removed via Percoll gradient centrifugation. Resulting cell bodies were incubated with anti-rat CD11b (i.e., microglia-specific) magnetic beads (Miltenyi Biotec #130–105–634; Auburn, CA) and then separated from the CD11b- population via magnetic columns. CD11b+ cell numbers were estimated on a hemocytometer, and 75,000 cells per well were incubated on 12 mm glass coverslips (Thermo Fisher Scientific #12–545–80) in media (1% L-glutamine, 1% Pen-Strep, 1% N2 media, 1% sodium pyruvate, 0.1% forskolin in DMEM) at 37 °C/5% $CO_2$ for 75 min with either vehicle, 60 ng (1x reconstitution; 200 μg/mL in PBS) neutrophil inhibitory factor (NIF), or 120 ng (2x reconstitution; 400 μg/mL in PBS) NIF (R&D Systems #5845-NF-050; Minneapolis, MN). Each condition was performed in duplicate. Fluorescbrite YG carboxylate microspheres (1μm diameter; Polysciences, Inc.; Warrington, PA) were added 1:1000 to each well and plates were re-incubated for 90 min. Immunocytochemistry was performed as in Derecki et al.[68] Coverslips were fixed with 4% PFA for 20 min at room temperature, and then washed in PBS. Coverslips were blocked and permeabilized for 1 h at room temperature (10% normal donkey serum, 0.5% BSA, 0.3% Triton x-100 in PBS), and then incubated with rabbit anti-Iba1 antibody (1:700; Wako Chemicals #019–19741; Richmond, VA) overnight at 4 °C. After washing in PBS, coverslips were incubated with donkey anti-rabbit AF 563 (1:500; Thermo Fisher Scientific) for 2 h at room temperature, washed, and mounted with Pro-Long Antifade Mounting media with DAPI (Thermo Fisher Scientific). Images comprised of DAPI, TexasRed (Iba1), and GFP (beads) were acquired at 20x magnification with a Zeiss AxioImager microscope equipped with a z-drive, Apotome optical dissector, and AxioCam HRm (Carl Zeiss Inc., Gottingen, Germany). Three 5-step (0.5 μm) z-stacks were acquired for each coverslip, and average z-projections were created using ImageJ (National Institutes of Health; Bethesda, MD).

For the NIF incubation timecourse, microglia from adult female brains ($n = 3$) were isolated, combined, and cultured as described above in media with 60 ng (1x reconstitution; 200 μg/mL in PBS) NIF for either 30, 60, or 120 min. Each condition was performed in duplicate. Coverslips were fixed with 4% PFA for 20 min at room temperature, and then washed in PBS. Coverslips were blocked and permeabilized for 1 h at room temperature (10% normal donkey serum, 0.5% BSA, 0.3% Triton x-100 in PBS), and then incubated sequentially, first with rabbit anti-6xHis antibody (1:700; Abcam ab9108; Cambridge, MA) then with mouse anti-CD11b (1:700;

EMD Millipore CBL1512; Burlington, MA), both overnight at 4 °C. After washing in PBS, coverslips were incubated with donkey anti-rabbit AF 488 and donkey anti-mouse 563 (1:500; Thermo Fisher Scientific) for 2 h at room temperature, washed, and mounted with Pro-Long Antifade Mounting media with DAPI (Thermo Fisher Scientific). Images comprised of DAPI, TexasRed (CD11b), and GFP (NIF) were acquired at 40x magnification with a Zeiss AxioImager microscope equipped with a z-drive, Apotome optical dissector, and AxioCam HRm (Carl Zeiss Inc., Gottingen, Germany). Three 7-step (0.3μm) z-stacks were acquired for each coverslip, and average z-projections were created using ImageJ (National Institutes of Health; Bethesda, MD).

**Stereotaxic microinjection.** Male (P30 or P54) and female (P22 or P54) rats were maintained under isofluorane anesthesia for the entire surgical procedure (2–3%; VetEquip; Livermore, CA). The scalp was cut midsagittally and Bregma was marked, after which two bilateral holes drilled at AP + 2.25 mm, ML ± 2.5 mm, DV −5.75 mm coordinates in P30 males, AP + 2.35 mm, ML ± 2.55 mm, DV -6.5 mm in P54 males, AP + 2.7 mm, ML ± 2.4 mm, DV -5.55 mm in P22 females, and AP + 2.4 mm, ML ± 2.45 mm, DV −6.05 mm in P54 females (Supplementary Fig. 2). A Hamilton syringe (Hamilton #7105; Reno, NV) was lowered to depth at a 10° angle and left in place for 1 min. NIF (1x reconstitution, 200 μg/mL NIF; R&D Systems; Minneapolis, MN) or vehicle (sterile PBS) was injected at a rate of 50 nL/min (60 ng in 300nL in P30 males, 70 ng in 350 nL in P54 males, 50 ng in 250 nL in P22 females, and 60 ng in 300 nL in P54 females). For scRNA and siRNA experiments, pre-validated sequences for rat (either control or D1r-specific) were purchased from ThermoFisher Scientific. The scRNA was a Silencer Select negative control (catalog #4390843), and the siRNA ID was s127671 (catalog #4390771). A final concentration of 200 μM sc/siRNA was co-injected with either NIF or vehicle at the same volumes listed above. The syringe was left in place for 5 min for diffusion, and then retracted and the procedure repeated on the other hemisphere. Saline was used to wet the scalp, and then the wound was closed with surgical staples and coated topically with Bupivicaine. An injection of Ketofen (5 mg/kg) was administered subcutaneously and the animal was placed in a clean cage with a food pellet and gelatinized water for recovery. After 2–3 h, the animal was re-paired with its cagemate and monitored for the next 48 h. Right versus left hemisphere and drug versus vehicle injections were counterbalanced, and the syringe was thoroughly cleaned with cleaning solution (Hamilton cleaning concentrate, Hamilton; Reno, NV) and distilled water before being used for the next surgery, or in the case of within-animal vehicle vs. NIF treatment, between injections. Injection coordinates and volumes were estimated with bromophenol blue dye in water and immediate decapitation, and an effective D1r siRNA dose was determined experimentally.

To perform the in vivo NIF timecourse, P54 females were unilaterally microinjected with NIF as described above, and tissue was collected 1, 3, and 5 days post-injection. Immunohistochemistry was performed as described above with anti-6x His tag antibody (Abcam #ab9108; 1:5000) and TSA amplification, and images were collected with confocal microscopy as described above with 40x magnification.

**Social behavior assessment.** Animals were handled on 3 occasions (~5 min per session) prior to any behavioral tests. Experimental animals were single-house for 24 h at age P21, P29, P37, or P53, while age-matched and sex-matched conspecifics were always group-housed (2–3 animals/cage). Social play tests were performed in the 3 h prior to the dark cycle (1600–1900) as follows: experimental and conspecific animals were each weighed and then the conspecific animal was placed in the experimental animal's home cage. Ten minutes of behavior was recorded using ANY-maze software (ANY-maze Behavioral tracking software; Wood Dale, IL), after which the conspecific animal was returned to its cage. When testing females, the experimental animal was inspected for a vaginal opening, and if appropriate, estrous smears were taken and later classified as metestrus, diestrus, proestrus, estrus phases of the estrous cycle according to Goldman et al.[69] Behavior initiated by the experimental animal over the first 9 min of the test was scored by a blinded investigator using Solomon Coder software (András Péter; solomoncoder. com). Social behaviors were determined following Manduca et al.[7]

(i) Percentage of time spent in social play, which included any classical play posturing (i.e., nape attacks, supine, and pins) in addition to pouncing on the conspecific, play boxing, and any rough-and-tumble play bouts.
(ii) Percentage of time spent in social exploration, which included grooming or sniffing the conspecific animal.
(iii) The number of nape attacks, supine positions, pin positions, and miscellaneous play elicitations (i.e., boxing, pouncing away from the neck, crawling over or under the conspecific).

Conspecific animals were never litter mates with the experimental animal and no more than +/−2 days different in age. Weight differences between the animals varied; however, difference in body weight did not correlate with total play scores (Supplementary Fig. 3) in males or females at any age.

**Statistical analysis.** Data points greater than two standard deviations above the group mean were excluded as outliers. All data were analyzed, and figures created with, GraphPad Prism 7 (GraphPad Software; La Jolla, CA). Data are depicted in

histograms as average ± standard error of the mean with individual data points overlaid. All statistics are located in Table 1 and Table S1.

Immunohistochemical analyses were performed as follows: the levels of all proteins assessed (D1r, C3, and Iba1) changed significantly over development (Fig. 1). Thus, increased microglial contact of these proteins could result from three mechanisms: (i) higher D1r (or C3/C3-D1r) levels might make microglial contact more likely due to the increased space occupied by D1rs; (ii) higher Iba1 levels may increase D1r contact by the same principle; or (iii) microglia may be actively contacting D1rs irrespective of fluctuating D1r or Iba1 levels. Each target protein (D1r or C3) was normalized for z-stack size, then masked volumes were calculated as a percentage of the total possible volume for each channel (i.e., masked D1r divided by total D1r), and divided by the percentage of Iba1 volume in the z-stack to account for changing levels of total Iba1. To assess association with CD68+ lysosomes, masked protein was calculated as a percentage of total possible volume. One-way ANOVAs were performed for all developmental time course data (i.e., P20, 30, 38, 54). In the event of significant ANOVAs, post-hoc Holm-Sidak t-tests were conducted (two-tailed). Because (a) immunofluorescent data needed to be acquired at different imaging parameters for males and females and (b) we were interested in sex differences in patterns of expression and behavior over development, and not sex differences at any one age, we did not incorporate sex as an independent variable, and rather performed a one-way ANOVA for each sex. For the analysis of D1r expression in NIF-treated and vehicle-treated NAc within the same animals, paired t-tests (two-tailed) were conducted.

For the phagocytosis assay, cell bodies were outlined by an experimenter blinded to the conditions using a merged DAPI and Iba1 image in the ROI manager of ImageJ. Densitometric measurements of the GFP channel (beads), were then acquired using the pre-set ROIs. One value (the average integrated GFP density for cells in each image) for each of 3 z-stacks, for each of 2 coverslips, for each of 4 animals, resulted in 24 data points. A one-way ANOVA was conducted, followed by Holm-Sidak post-hoc t-tests (two tailed). For the NIF timecourse assay, densitometric measurements of the GFP and TxRed channels were acquired and normalized for number of cells present in the image. One value per channel (the average integrated density for each image) for each of 3 z-stacks, for each of 2 coverslips, resulted in 6 data points per condition. A one-way ANOVA was conducted for both the GFP (NIF) and TxRed (CD11b) data.

For behavioral analyses, duration of social play and social exploration were calculated as a percent of the total testing period (500–540 s), frequency of play postures was calculated for nape attacks, supine, pin, miscellaneous play elicitation, and total play actions, and then one-way ANOVAs were conducted separately for males and females. When appropriate, Holm-Sidak post-hoc t-tests (two tailed) were conducted. In bilateral NIF- and vehicle-treated animals, behavior was analyzed with unpaired t-tests (two-tailed). Pearson's r was calculated to determine if change in weight from conspecific or estrous stage was significantly correlated with social behaviors.

## Data availability

The data that support the findings of this study are available from the authors on reasonable request.

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

## Acknowledgements
We thank Phuong K. Tran and Ernesto Barbosa for help with experiments, and members of the Bilbo lab for helpful discussion during the development of this project. Additionally, we thank the Caron lab at Duke University and the Lobo lab at University of Maryland Medical School for generously sharing mouse brain tissue for antibody validations. This work was supported by RO1 DA034185 and RO1 MH101183 to SDB and F32DA043308 to AMK.

## Author contributions
A.M.K., C.J.S. and S.D.B. planned the experiments. A.M.K., C.J.S., N.R.A., and S.C.S. performed experiments. A.M.K. analyzed the data and wrote the manuscript. All authors contributed to manuscript edits.

## Additional information

**Competing interests:** The authors declare no competing interests.

