## [Peer Review File · Nature Communications]

Reviewers' comments:

Reviewer #1 (Remarks to the Author):

This manuscript by Kopec et al., outlines the interaction between microglia and D1 receptors during adolescence. The manuscript shows a reduction in D1 receptors that develops over the adolescence period that the authors argue regulates play behavior. This process was shown to be different between males and females. They provide a potential mechanism for these effects highlighting the ability of complement signaling to alter discrete aspects of social behavior in males only. The manuscript is extremely well written. While this manuscript is interesting and the findings are important, the studies in their current form are correlative and preliminary. Thus, the work may be better suited for a more specialized journal at this time. Below I have listed my specific comments.

1. A major concern is the use of D1 receptor antibodies. In the past, these have been shown to be notoriously non-specific and have cross-reactivity with other dopamine receptors as well as other signaling molecules. Because the entire manuscript relies on immunofluorescence, validation for these antibodies should be shown. This validation should be done in brain.

2. Along a similar line, immunofluorescence is not the best way to show quantitative changes in receptor levels over time. The authors should show that this occurs with western blot, which is more accurate for this question.

3. In the figures the authors present data from immunofluorescent quantification. However, all of the data is normalized to D1 levels. The authors should show the pattern of C3 not normalized to D1 levels as well. This is important for evaluating the findings in the manuscript.

4. Why in the representative images in figure 3 does it look like there is more D1 in females, when in the previous figure it looks like D1 is undetectable in females. Are these images actually representative?

5. A large majority of the work is correlative. The authors argue that C3 mediated internalization is driving the behaviors; however, enhanced pruning of neurons in general would like result in a decrease in marks of those neuronal subtypes. Thus, it is not surprising that C3 is correlated with reductions in D1. The authors need to show that the effect of C3 on behavior is in fact directly mediated by changes in D1 receptors to make the strong conclusions stated within the manuscript. Below are a number of things that should be addressed:

a. It is critical to add the behavioral data with NIF infusions in females. Why was this not included originally? It seems critical to the conclusions of the manuscript.

b. It will be important to do the time-course from figure 1 with NIF infusions.

c. Validation that this effect is mediated via synaptic pruning and not via changes in D1 translation should be included. The experiments are conducted 2-4 hours post infusion. Is this enough time that blocking synaptic pruning would increase D1 receptor protein? A time

course should be done to show that over that 2-4 hour window that there would typically be a reduction in D1 receptors and that NIF is blocking this reduction. If that does not occur, it would seem that NIF is not blocking pruning, but rather upregulating D1 receptors.

6. The authors should determine and comment on why this sex difference is occurring. Are C3 receptors much less in females? It is surprising that the drug, NIF, does not induce an effect in females. This may suggest that the receptor or associated signaling may not be present.

Reviewer #2 (Remarks to the Author):

The aim of this study was to test the involvement of microglia and the complement system in the elimination of NAc dopamine D1 receptors that occurs during adolescence, and to determine whether this microglia-mediated refinement is responsible of adolescent-specific behavioral changes in rats.

Few studies have reported that microglia have a critical role in pruning synapses during development (Paolicelli et al, Science 2011; Schafer et al., Neuron 2012). For instance, it has been shown that juvenile mice lacking the chemokine receptor Cx3cr1, that it is essential for microglia migration, show reduced synaptic pruning during development associated with persistent decrease in brain connectivity and with deficits in social interaction (Zhan et al., Nat Neurosci 2014). Thus, the notion that microglia-mediated synaptic pruning is developmentally relevant and may be required for behavioral changes occurring during development, particularly in the social context, is nowadays accepted. The present study is very interesting, since it confirms and extends this scenario by showing a microglia-regulated refinement of D1 receptors in the rat NAc, that may account for developmental changes in social behavior. I have the following comments:

1. A limitation of the behavioral approach used is that the authors analyzed and reported behavior as percent time spent either doing social play or social exploration. It would be far more useful to know which exact behavior is really affected. Is it the pinning, or play solicitation, or evasion? This is particularly important. Indeed, it is known that, in rats, social play peaks in-between postnatal days 28–40, and declines thereafter as the animals become sexually mature. However, not only the frequency, but also the structure of social play behavior in rats changes during development (Pellis and Pellis, Aggr Behav 1987; Pellis and Pellis, Dev Psychobiol 1990). Whereas play solicitation does not change in terms of structure (i.e., pouncing), but rather in frequency, the responses evoked by play solicitation do change. Thus, at its onset, standing defense is the most common response, which around weaning (i.e., around postnatal days 21) is replaced by pinning as the most likely response. With the onset of puberty, however, rotating to supine declines, so that standing defense becomes the most widely used response. This latter transition, however, occurs only in male rats. Therefore, since the authors claim sex-specific behavioral changes, it is important to determine which behavioral pattern actually differ between males and females in the course of development, and how they are affected by microglia-mediated remodeling of NAc circuit. A more refined behavioral analysis may also allow to appreciate sex-specific behavioral changes in social play. It is indeed odd that, as the results are presented now,

the authors did not observe more social play in males rather than in females, with the only difference maybe being detected on postnatal day (PND) 30.

2. Another important issue to consider is that, in the present study, experimental animals were singly housed for 24 h before testing, while stimulus animals were always group housed. Since 0 or 24 h of isolation before testing induce minimal and maximal levels of social play behavior, respectively (Niesink and Van Ree, *Neuropharmacology* 1989; Vanderschuren et al, *Psychopharmacology* 1995), the two animals of the pair were not equally motivated to play. Since social play is a reciprocal social activity, that is highly influenced by the behavior displayed by the test partner, it would be more appropriate to pair animals equally motivated to the social interaction.

3. Could the NIF-induced reduction of social play behavior be non-specific? While the fact that general social exploration was unaffected by NIF infusion argues against this possibility, it would be useful to report whether NIF affected locomotor activity during the social encounter. Again, it would be important to know which actual behavioral parameter was affected by NIF.

4. Concerning the results shown in Figure 4B, it is unclear whether the same animals were tested over repeated days, or whether different cohorts of rats were used to study social play at each age.

5. The authors show that NIF injection increased social play in male rats at PND 38 (Fig. 5B). However, in this experiment, control animals exhibited a baseline level of social play that is approximatively half of that displayed by control animals at PND 38 in Figure 4B1. Again, showing the individual behaviors actually displayed during social play may help to resolve this discrepancy.

Minor comments:

The authors should mention in the abstract and in the introduction that the study was performed in rats

Reviewer #3 (Remarks to the Author):

This study by Kopec et al. addresses the question of how microglia regulate development of the dopaminergic system through adolescence, and how these immune cells may influence social behavior. They focus specifically on regulation of NAc D1 receptors, and find a sex-specific effect in which D1 receptors are eliminated by phagocytosis in males but not females during adolescence. They report that this microglia-mediated phagocytosis is required for normal adolescent social play behavior in males. This study addresses interesting questions (how reward circuitry underlying social behavior changes through adolescence; sex differences in these developmental effects), although the motivation for specifically looking at the involvement of microglia is not clear, and there is a lack of substantial causal data to support their conclusions. The majority of the experiments are

purely correlative in nature, showing similar time courses of C3 expression, D1r expression, and social play behavior, with limited causal evidence for a direct link between microglia activation, D1 receptor elimination, and social behavior. More in vivo work is needed to flesh out the causal behavioral effects of manipulating each of these molecular mechanisms. Detailed comments are listed below:

Line 94: The authors state that “microglia... are critical mediators of neural circuit development” – but is this known to still be true in adolescence vs prenatal development? Generally the background/rationale for looking at microglia in the adolescent dopaminergic system is lacking.

What type of validation has been done to show the specificity of their D1r antibody? Historically, D1 and D2 have not been easy to stain for—how do we know that the antibodies used here are selective? For example, have they shown overlap with a validated D1-Cre line, or with dynorphin staining (and conversely no colocalization in a D2-Cre/A2a-Cre line or with enkephalin staining)?

In Figure 1, there doesn't seem to be much added benefit is of looking at C3-D1r colocalization (as opposed to C3 and D1r expression individually). With this degree of resolution, colocalization doesn't say anything about actual interaction between the two at the synaptic level or seem to add much beyond simply describing the time course of expression of each individually.

Line 151: “...an iC3b-D1r tagging mechanism resulting in D1r elimination may be employed in male adolescent NAc development.” The authors need to be careful about overstating their claims—here, the similar time course of iC3b-D1r co-IP and D1r downregulation is just correlative, and this should be worded conservatively, being careful not draw a causal conclusion.

Line 166: “In males, both D1rs (Fig. 2B1; Table 1I) and C3-D1rs (Fig. 2B3; Table 1K) are, indeed, maximally contacted by microglia at P30 prior to D1r downregulation” But in Fig 2B3 it looks like C3-D1rs are maximally contacted at P38, not P30?

Generally throughout the figures, panel sub-labeling is in an unusual format (e.g. Fig. 2B1, 2B2) -- could be replaced by giving each panel/plot a unique letter.

Figure 3A is still looking again at purely correlative data – it is not very compelling to keep showing general temporal correlation of expression of various microglial markers. Some of this correlative data could be moved into a supplementary figure in exchange for including more causal data in the main figures. It is only at the end of Figure 3, more than halfway through the paper, that the data begins to show something causal– i.e. blocking C3 binding and phagocytosis with NIF in Fig 3E,F.

Generally, the legends and images should explicitly state more clearly what is being labeled by each color (e.g. Figure 3E,F nowhere on the figure or in the legend does it state that green represents, presumably, D1r staining).

In Figure 3, the authors inject NIF at P30 and measure D1r at P38. How long after in vivo injection does NIF stay in the NAc? How soon after NIF injection would D1r expression go up? More details on the in vivo action of NIF are needed.

Figures 4 and 5 could easily be combined into a single figure.

Similarly, more experiments are needed to flesh out Figure 5. What effect does NIF injection have on female social behavior? What if you inject NIF at a time point other than P30 in males? Is this effect really specific to that particular developmental timepoint or would NIF injection recapitulate this behavioral effect at any age?

While Figure 5 addresses the causal role of microglia in social behavior, these experiments do not demonstrate that the behavioral effect of NIF is mediated by D1 receptors. This is a key lacking piece of evidence. More in vivo experiments are needed to more tightly link each of these elements: microglia, D1 receptors, and social behavior.

Generally, how specific are the observed effects (behavioral and histological) to NAc? Do D1 receptors in other brain regions show the same pattern?

Reviewer #4 (Remarks to the Author):

The authors present a potentially interesting and provocative story on the involvement of microglia and complement in D1R elimination in nucleus accumbens. There is substantial potential novelty as this would be a report that a role for the complement cascade in the CNS is sex dependent. Moreover, the effects of microglia/complement are much later in development than previously reported in other systems. Thus, there would be considerable interest in this story. However, there are serious deficiencies in the evidence supporting the main conclusions of the paper, and inconsistencies in parts of the story, that preclude me from endorsing the present manuscript.

A main concern is that the manuscript is comprised of two distinct and only weakly related parts – the immunocytochemical/colocalization part and the in vivo behavioral part. This is particularly so because the amount of work in vivo with NIF appears to be a single experiment, in males only, and with only a vehicle control. The story developed in the first part of the paper is that there is a sex difference in loss of D1Rs. The authors need to test whether there is, is not, a sex difference in the effects of NIF. Ideally, there would be a scrambled or other control for NIF. At least they could take advantage of the U-shaped NIF dose-response relationship and give higher doses of NIF in vivo which would be predicted to not prevent loss of D1Rs and not to 'rescue' the behavior. Also, to more rigorously interrogate their hypothesis the authors need to explore possible off-target effects by delivering NIF at times when the first part of the paper would suggest no involvement of complement (ie. P54 in males, or any time in females). Even so, the first part of the paper is about D1Rs and there is no evidence (or even test) of involvement of D1Rs in the behavioral change shown in Fig. 5B. The authors need, again to better interrogate their

hypothesis, which would predict that the increase in total play time is dependent upon increased levels of D1Rs. This prediction needs to be tested experimentally. Otherwise there is only a weak, circumstantial link between first and second parts of the paper. To the extent that the NIF experiments with D1R immunofluorescent imply the blocking complement-mediated phagocytosis blocks the drop in D1Rs in males, why do D1R levels not increase again at P54 when phagocytosis seems to be lacking.

To further support the necessity of the cells and pathways they are investigating the authors need to have more than just a single reagent, NIF. They should try interventions to eliminate or silence microglia, or animals in which key components of the pathways have been eliminated genetically. Either of these would greatly strengthen the conclusions of the paper.

There are important concerns with the co-IP experiments. If D1Rs are indeed 'tagged' by iC3b then, from previous literature on this highly reactive fragment, this should form a covalent ester bond, that would not be broken even under boiling in SDS, and should see a complex of iC3b-D1R which would run at a higher molecular weight than that shown. Thus, seeing C3 immunoreactivity at the low molecular weight shown is very curious and not consistent with the authors' interpretation. They should show the entire blot to see if there are bands at the correct predicted size. Also, to fully support the conclusions the experiments should do reciprocal experiments to determine whether IPing C3 (iC3b) can co-IP D1R. In addition, the authors present no evidence for quantification of the results shown in the images, nor whether the images are representative of replicates or single experiments. Finally, they need to provide a credible explanation for the high iC3b signal in the coIP at P54F, when both the D1R and C3 levels are at their lowest.

Other concerns

There is substantial inconsistency in the images in the panels in Fig 1. As one example, in panel B P30 the maximum C3 volume should be about 0.5% whereas the average for what should be shown in panel F P38 should be about 0.4%. Yet the image in B is massively brighter than that in F. Likewise, C P30 should have only about 20% more immunofluorescence than G P20, which is not consistent with the images. This dramatic inconsistency undercuts the reliability of the data shown in this and all immunofluorescence figures.

The authors regularly refer to phagocytic elimination of D1Rs. What do they mean by this – that the receptors are somehow plucked from the neuronal membrane, that substantial chunks are taken from the neurons, or something else?

The authors find that NIF doesn't completely eliminate phagocytosis. It would be interesting to compare effect of NIF to that of Cd11b KO to determine how much of the Cd11b-dependent phagocytosis is blocked.

The other measures the authors saw correlate with D1R loss (increased C3-D1R microglia contact and/or increase D1R in CD68 lysosomes) should be examined with NIF as further confirmation that the drug is acting on this phagocytosis pathway.

A point which seems to be ignored by the authors is that the D1R volume in males is much lower than that in females. What is the explanation for that in the context of the present study?

Given the NIF dose response in Fig 3d, how did the authors select the dose to be used in vivo, given that the NIF is differentially diluted when injected in vivo as compared with when applied in the bathing solution in vitro?

In many places in the manuscript the writing is excessively leading, presuming the conclusions before doing the experiments.

Response to Reviewers

We thank the reviewers for their thorough analysis; the manuscript has been significantly strengthened by virtue of their helpful comments. Please note that figure organization has changed substantially based on reviewer comments and the incorporation of new data. Major new experimental additions include re-analysis of all behavior videos to score individual components of play behavior and several new causal experiments, including P22 female manipulations, P22 female and P30 male D1r siRNA manipulations, and P54 manipulations in both sexes. Below we respond point-for-point to the reviewers' critiques.

Reviewer #1

1. A major concern is the use of D1 receptor antibodies. In the past, these have been shown to be notoriously non-specific and have cross-reactivity with other dopamine receptors as well as other signaling molecules. Because the entire manuscript relies on immunofluorescence, validation for these antibodies should be shown. This validation should be done in brain.

We have completed an identical IHC procedure as reported in the Methods on D1-TdTomato mouse brains, and have included an image demonstrating extensive overlap in the TdTomato and D1r immunofluorescent signals in Fig. S1A.

2. Along a similar line, immunofluorescence is not the best way to show quantitative changes in receptor levels over time. The authors should show that this occurs with western blot, which is more accurate for this question.

Thank you for this suggestion. We originally planned to perform Western blotting to assess D1r and C3 levels in the NAc. However, we learned after performing pilot IHC analyses that the region of NAc that is enriched for D1rs is quite discrete (at least at the ages we assessed), specifically the dorsomedial quadrant of the NAc (see Fig. S1B). Given this, we are not confident we could consistently dissect our region of interest for homogenization and blotting. Rather, using D1r immunoreactivity permits a more controlled and precise way for us to consistently assess the same regions in both sexes and across ages. A lack of precision in this regard is much more likely to produce false positive or negative data. Our IHC analyses were carefully performed such that all tissue to be directly compared were processed at the same time and imaged with the same parameters. Your concern is precisely why we did not make statistical comparisons between the sexes – the tissue was processed at different times and needed different image acquisition parameters. Importantly, quantitative measurements have been taken with Imaris volumetric reconstructions (Schafer et al. 2012; Weinhard et al. 2018) and the protocol on which we based our own analyses has been published (Schafer et al. 2014).

3. In the figures the authors present data from immunofluorescent quantification. However, all of the data is normalized to D1 levels. The authors should show the pattern of C3 not normalized to D1 levels as well. This is important for evaluating the findings in the manuscript.

We apologize that our methods were unclear. Those data were not all normalized to D1r levels. First, total D1r, C3, and Iba1 were normalized to the ROI volume, to account for changes in z-stack thickness. The normalization to D1r levels to which you are referring was used as an example to explain how masked D1r data (i.e. the volume of D1r in contact with microglia (Iba1)) was normalized to total D1r volume within the same ROI. The logic here was to ensure that changes in microglial contact of D1rs, in this example, was not just due to changing levels of D1r in the tissue. We have moved this example calculation to the Methods so there is no confusion while reading the results as to this important point.

4. Why in the representative images in figure 3 does it look like there is more D1 in females, when in

the previous figure it looks like D1r is undetectable in females. Are these images actually representative?

Thank you for this comment. D1r levels in male and female tissue needed to be imaged at separate acquisition parameters, so neither the images themselves nor the data point values can be directly compared between the sexes. In Fig. 3, in addition to being imaged at different acquisition parameters, the data represent age P38 in males and P30 in females. With that said, we regret there was an oversight in the selection of the male representative images for Fig. 3: while those images were true data, they did not adequately capture the *average* change in D1r levels resulting from NIF injection (relative to vehicle injection in the contralateral hemisphere). Those images have been replaced with images more representative of the average data. We also re-checked that the images in Fig. 1 best represented the average data, and all of those images were retained. To alleviate these concerns, we have (i) stated more explicitly in the results and figure legends that male and female data cannot be directly compared (lines 132-134; 791-793), (ii) replaced the male Vehicle:NIF images in Fig. 2F, (iii) added timelines to figures to make the stereotaxic data more easily understood, (iv) re-cast the within-animal data histograms to represent % change from within-animal vehicle control so the relative change in D1r immunoreactivity is clearer (Fig. 2F,H), and (v) re-built our figures in Adobe Illustrator in an attempt to attain better quality images.

A large majority of the work is correlative. The authors argue that C3 mediated internalization is driving the behaviors; however, enhanced pruning of neurons in general would like result in a decrease in marks of those neuronal subtypes. Thus, it is not surprising that C3 is correlated with reductions in D1. The authors need to show that the effect of C3 on behavior is in fact directly mediated by changes in D1 receptors to make the strong conclusions stated within the manuscript. Below are a number of things that should be addressed:

5. It is critical to add the behavioral data with NIF infusions in females. Why was this not included originally? It seems critical to the conclusions of the manuscript.

We did not include female NIF infusions in the original manuscript because we had no evidence to suggest that the C3-microglial phagocytosis process was occurring in females to mediate D1r levels. We have now included NIF injections into the NAc in females at an age prior to their D1r decline (P22) and examined their social behavior at P30, with the expectation that we would see no change in social behaviors at P30, as C3 – C3R interactions don't impact developmentally-typical D1r downregulation in IHC analyses (Fig. 2). To our surprise, NIF also increased social play (but not social exploration) in females (Fig. 3), and importantly, this effect does not appear to be mediated by D1rs (Fig. 4). We think these experiments may have uncovered a fundamental difference in the utilization of C3 – C3R interactions between the sexes: in males, this process mediates overt D1r elimination to decrease social play over the course of adolescence; in females, this process mediates a basal/homeostatic environment in the NAc, through regulation of as yet unknown receptor/synapse components, to regulate basal levels of social play. We expand on this theory further in response to one of your subsequent comments (#8) as well as in lines 329-347.

6. It will be important to do the time-course from figure 1 with NIF infusions.

We agree that doing a full timecourse would lend specificity to our results. Therefore, we now add P54 NIF/Vehicle manipulations in both sexes to explore the specificity of NIF treatment to developmental processes, as we expect most development in this region to be in progress if not completed by P54. The data in Fig. S5 demonstrate no effect of NIF on social behaviors at this age in either sex. We did not conduct a full timecourse, because we do not have *a priori* hypotheses as to the effects of NIF treatment at ages other than the ones tested herein (P22 in females, P30 in males, P54 in both sexes). These will be interesting manipulations to characterize more fully in future experiments.

7. Validation that this effect is mediated via synaptic pruning and not via changes in D1 translation should be included. The experiments are conducted 2-4 hours post infusion. Is this enough time that blocking synaptic pruning would increase D1 receptor protein? A time course should be done to show that over that 2-4 hour window that there would typically be a reduction in D1 receptors and that NIF is blocking this reduction. If that does not occur, it would seem that NIF is not blocking pruning, but rather upregulating D1 receptors.

We apologize for the confusion. NIF injections were completed **8 days** prior to behavioral testing or immunohistochemical assessment. While a very important point, the data in Fig. 2 demonstrating augmented D1r levels in response to NIF infusion (8 days prior) in males, but not females, would suggest that the effect of NIF cannot be merely to increase D1r translation, or one would expect to see this in both sexes. In addition to adding timelines to clarify when injections were performed and tissue collected (Figs. 2E,G), we have added discussion of your important concern in lines 357-359.

8. The authors should determine and comment on why this sex difference is occurring. Are C3 receptors much less in females? It is surprising that the drug, NIF, does not induce an effect in females. This may suggest that the receptor or associated signaling may not be present.

Excellent question, and one we are puzzling over ourselves! The new data have added an additional layer of complexity: NIF *is* effective in mediating social play in females at P22, but not via D1r regulation (Fig. 4H). We interpret our data as a whole to suggest two novel concepts: first, C3-microglia interactions are regulating the molecular environment of the NAc in male and female rats in different ways. Specifically, in males this immune process regulates D1r levels, resulting in overt developmental changes in social play behavior. In females, we predict C3 – C3R interactions may play more of a homeostatic, maintenance mechanism, potentially regulating basal levels of other receptors important for social behavior (e.g., opioid receptors). We predict this because NIF was able to increase social play (albeit modestly given the levels of play are lower in females) in females despite no clear changes in social play behavior in *natural* development (Fig. 3). We hypothesize that interfering with C3-microglial processes with NIF somehow created an imbalance in the NAc in females, thus revealing changes in play behavior that would not normally be observed. This is speculative, but we are excited to follow these hypotheses in future experiments. We discuss these ideas in lines 329-347.

Reviewer #2

9. A limitation of the behavioral approach used is that the authors analyzed and reported behavior as percent time spent either doing social play or social exploration. It would be far more useful to know which exact behavior is really affected. Is it the pinning, or play solicitation, or evasion? This is particularly important. Indeed, it is known that, in rats, social play peaks in-between postnatal days 28–40, and declines thereafter as the animals become sexually mature. However, not only the frequency, but also the structure of social play behavior in rats changes during development (Pellis and Pellis, *Aggr Behav* 1987; Pellis and Pellis, *Dev Psychobiol* 1990). Whereas play solicitation does not change in terms of structure (i.e., pouncing), but rather in frequency, the responses evoked by play solicitation do change. Thus, at its onset, standing defense is the most common response, which around weaning (i.e., around postnatal days 21) is replaced by pinning as the most likely response. With the onset of puberty, however, rotating to supine declines, so that standing defense becomes the most widely used response. This latter transition, however, occurs only in male rats. Therefore, since the authors claim sex-specific behavioral changes, it is important to determine which behavioral pattern actually differ between males and females in the course of development, and how they are affected by microglia-mediated remodeling of NAc circuit. A more refined behavioral analysis may also allow to appreciate sex-specific behavioral changes in social play. It is

indeed odd that, as the results are presented now, the authors did not observe more social play in males rather than in females, with the only difference maybe being detected on postnatal day (PND) 30.

We thank you for your expertise and thorough assessment of our behavioral data. We agree with you that the details of the social play encounter are important and highly informative and apologize for their initial omission. Therefore, according to your comments, we have re-scored our behavioral videos to assess the number of nape attacks, pinning/supine positions, and miscellaneous play elicitation (defined as boxing, climbing on top of or beneath, or attacks targeted away from the neck; Figs. S3,4,6). While no one aspect/component of play continuously predicted increased overall time spent in play, there were often increased nape attacks and supine positions in groups with greater overall play.

With regards to sex differences in social play, previous work suggests that whether or not sex differences are observed depends on the behavioral paradigm (Pellis & Pellis, 1990). Specifically, in behavioral paradigms in which social play is tested among cage mates and under undisturbed conditions, sex differences are observed (Meaney & Stewart, 1979, 1981; Poole & Fish, 1976; Parent & Meaney, 2008). However, in behavioral paradigms in which animals are briefly isolated and then tested in pairs (used here), no sex differences in social play are observed (Panksepp & Beatty, 1980; Panksepp, 1981; Thor & Holloway, 1984; Veenema et al., 2013; Bredewold et al., 2014, 2015). Indeed, one study has even found that play soliciting behaviors were higher in females than in males when playing with an unfamiliar partner (Circulli et al., 1996). Finally, studies have shown that even in instances in which play does not differ between males and females, the neural mechanisms by which social play is regulated do differ between the sexes (Bredewold et al., 2014, 2015). We thus expect that sex differences may very well be present if play was assessed in a different behavioral paradigm than what we have chosen to use.

10. Another important issue to consider is that, in the present study, experimental animals were singly housed for 24 h before testing, while stimulus animals were always group housed. Since 0 or 24 h of isolation before testing induce minimal and maximal levels of social play behavior, respectively (Niesink and Van Ree, Neuropharmacology 1989; Vanderschuren et al, Psychopharmacology 1995), the two animals of the pair were not equally motivated to play. Since social play is a reciprocal social activity, that is highly influenced by the behavior displayed by the test partner, it would be more appropriate to pair animals equally motivated to the social interaction.

Thank you for raising this important point. In line with your comments, we specifically chose to isolate the focal/subject animal for 24 hours prior to social play testing because this procedure has been shown to maximize acute levels of social play behavior during the testing period (Panksepp & Beatty, 1980; Niesink & Van Ree, 1989; Vanderschuren et al., 1995). In our experiments we intentionally sought to create a disparity between the social motivation of our subject animal (which received all pharmacological manipulations) and the stimulus animal (not pharmacologically targeted). To do this, we utilized the resident-intruder social play test (Thor and Holloway, 1984). Because we were performing pharmacological manipulations, we wanted to use a design that would allow us to isolate the effects of our drug administration on the subjects' motivation/behavior with minimal interference from the behavior of the stimulus animal. This is a well characterized behavioral paradigm that has been used extensively in the context of pharmacological interventions and social play behavior (Thor & Holloway, 1984; Veenema et al., 2013; Bredewold et al., 2014, 2015; Reppucci et al., 2018).

11. Could the NIF-induced reduction of social play behavior be non-specific? While the fact that general social exploration was unaffected by NIF infusion argues against this possibility, it would be useful to report whether NIF affected locomotor activity during the social encounter. Again, it would be important to know which actual behavioral parameter was affected by NIF.

This is a very good question, and we have performed a group of NIF or vehicle injections in P54 animals, when we expect these developmental processes to be in progress or complete in both sexes, to address the specificity of these manipulations. We observed no social behavior changes in these animals, see Fig. S5. In accordance with your observation that while NIF changes play, it does not change other aspects of social behavior, we have added more NIF-treated groups (females in particular), and in no instance does social exploration change – it is always a specific effect on social play (Fig. 3-4). Moreover, in P54 manipulations there is no change in social play in either sex, suggesting that the effect of NIF is not to increase overall hyperactivity. We see this report as an important first step in exploring this interesting phenomenon.

12. Concerning the results shown in Figure 4B, it is unclear whether the same animals were tested over repeated days, or whether different cohorts of rats were used to study social play at each age.

Thank you; animals were tested only once. We have clarified this in lines 218-221, and have added a timeline to make this clear within the figure (Fig. 3A).

13. The authors show that NIF injection increased social play in male rats at PND 38 (Fig. 5B). However, in this experiment, control animals exhibited a baseline level of social play that is approximately half of that displayed by control animals at PND 38 in Figure 4B1. Again, showing the individual behaviors actually displayed during social play may help to resolve this discrepancy.

Excellent point. The data suggest that surgeries themselves decreased play behavior writ large. A deep ventral injection such as those required for NAc manipulations can certainly disrupt neural circuits in other regions, making the vehicle control groups essential for data interpretation. We have now analyzed the individual components of play as you suggest, which can be found in Fig. S4,6. While the number of play actions were also dampened in vehicle treated animals in some surgical groups (Fig. S4), this was not always the case (Fig. S6), so we think that while surgical intervention can certainly have a major impact on behavior, inter-animal and inter-cohort variability is also playing a role in these data. To that end, it is encouraging to see the NIF effect replicated in the new behavioral groups (Fig. 4).

14. The authors should mention in the abstract and in the introduction that the study was performed in rats

We have included the model organism in the abstract, introduction, and title.

Reviewer #3

15. Line 94: The authors state that “microglia... are critical mediators of neural circuit development” – but is this known to still be true in adolescence vs prenatal development? Generally the background rationale for looking at microglia in the adolescent dopaminergic system is lacking.

Thank you for identifying an area that needed clarification. The role of microglia in synaptic development, particularly for their role in the elimination or remodeling of synapses, has been reported in several brain regions (Schafer et al. 2012; Stevens et al. 2007; Paolicelli et al. 2011; Weinhard et al. 2018). Interestingly, previous work also suggests that an immune mechanism engaged in the NAc during adolescence, but not young adulthood (i.e. P38 vs. P54), mediates later life addiction-related behaviors (Schwarz & Bilbo 2013), indicating that there is a sensitive window of time in which immune mechanisms in the NAc can affect neural maturation and behavior. Because of these reports and the well-documented dopaminergic development that takes place in the NAc and other regions of the reward circuitry during adolescence, we asked whether similar microglia-mediated developmental mechanisms could be at play during this phase of brain development. Our report is the first to our knowledge to report this same kind of developmental process is mediating sex-specific dopaminergic development in the NAc during adolescence, and also the first to report that normal developmental

changes in behavior are contingent on these immune processes. We have clarified this rationale (lines 87-111).

16. What type of validation has been done to show the specificity of their D1r antibody? Historically, D1 and D2 have not been easy to stain for—how do we know that the antibodies used here are selective? For example, have they shown overlap with a validated D1-Cre line, or with dynorphin staining (and conversely no colocalization in a D2-Cre/A2a-Cre line or with enkephalin staining)?

This is a very important point, and we now include an image of our IHC protocol for D1r in D1-TdTomato mouse brain (Fig. S1A). We find that this antibody is extremely sensitive to epitope retrieval techniques (detailed in the Methods) which may explain historically poor staining.

17. In Figure 1, there doesn't seem to be much added benefit is of looking at C3-D1r colocalization (as opposed to C3 and D1r expression individually). With this degree of resolution, colocalization doesn't say anything about actual interaction between the two at the synaptic level or seem to add much beyond simply describing the time course of expression of each individually.

We agree completely; we have removed these data from Fig. 1 and now introduce the colocalized C3-D1r signaling as a tool for probing microglial contact and lysosomal content, rather than a discrete point of data itself (lines 147-149).

18. Line 151: "...an iC3b-D1r tagging mechanism resulting in D1r elimination may be employed in male adolescent NAc development." The authors need to be careful about overstating their claims—here, the similar time course of iC3b-D1r co-IP and D1r downregulation is just correlative, and this should be worded conservatively, being careful not draw a causal conclusion.

We thank the reviewer for catching this and have changed the language to be more conservative.

19. Line 166: "In males, both D1rs (Fig. 2B1; Table 1I) and C3-D1rs (Fig. 2B3; Table 1K) are, indeed, maximally contacted by microglia at P30 prior to D1r downregulation" But in Fig 2B3 it looks like C3-D1rs are maximally contacted at P38, not P30?

You are quite right. While not being significantly different from P30, contact at P38 is higher. We have amended our language.

20. Generally throughout the figures, panel sub-labeling is in an unusual format (e.g. Fig. 2B1, 2B2) -- could be replaced by giving each panel/plot a unique letter.

Thank you for the suggestion; we have changed the figure labeling throughout.

21. Figure 3A is still looking again at purely correlative data – it is not very compelling to keep showing general temporal correlation of expression of various microglial markers. Some of this correlative data could be moved into a supplementary figure in exchange for including more causal data in the main figures. It is only at the end of Figure 3, more than halfway through the paper, that the data begins to show something causal– i.e. blocking C3 binding and phagocytosis with NIF in Fig 3E,F.

Thank you for this suggestion. We have now combined the correlative data into one figure (Fig. 1).

22. Generally, the legends and images should explicitly state more clearly what is being labeled by each color (e.g. Figure 3E,F nowhere on the figure or in the legend does it state that green represents, presumably, D1r staining).

Thank you, we have more carefully labeled figures throughout.

23. In Figure 3, the authors inject NIF at P30 and measure D1r at P38. How long after in vivo injection

does NIF stay in the NAc? How soon after NIF injection would D1r expression go up? More details on the in vivo action of NIF are needed.

These are very important questions. NIF is a very well-characterized peptide: NIF binds to the I domain of the CD11b subunit of C3 receptor (but not CD11a or CD11c; Moyle et al. 1994; Muchowski et al. 1994), and pulse-chase experiments suggest the binding is at a high enough affinity for originally bound NIF to not be displaced by infusion of new NIF in *in vitro* conditions with neutrophils; moreover, the reversibility of NIF binding to neutrophils appeared to be dependent on the activation state of the cell (Moyle et al. 1994). NIF occludes the ability of other CD11b ligands, including iC3b (the enzymatically cleaved version of C3 thought to act as the 'tag') and fibrinogen (Muchowski et al. 1994; Zhang & Plow 1996), but not Factor X (Muchowski et al. 1994). Extensive studies have been performed to characterize its binding to CD11b, even identifying a critical series of amino acid residues required for binding: Asp¹⁴⁹, Arg¹⁵¹, Gly²⁰⁷, Try²⁵², and Glu²⁵⁸ (Ustinov & Plow 2002). In human and dog studies, pharmacokinetic modeling suggests that UK-279,276 (recombinant NIF approved for human treatment) is cleared through both receptor binding and subsequent subtle dose-dependent decreases in the overall volume of UK-279,276 as well as elimination putatively through CD11b-NIF degradation (Jonsson et al. 2005; Webster et al. 2005); conversely there appears to be a more simplified, single clearance process in rats (Webster et al. 2005). UK-279,276 affinity is also higher for CD11b in dogs and humans (IC₅₀=<0.5nM) than in rats (IC₅₀=134nM; Webster et al. 2005). A clinical trial of a single NIF infusion from 0.06-1.5mg/kg in stroke patients was well tolerated at all doses with no evidence of adverse side effects (Lees et al. 2003). At the 1.0 and 1.5mg/kg doses, CD11b saturation was >80% for at least 7 days after treatment (Lees et al. 2003).

Our report will be the first, to our knowledge, to infuse NIF locally into the brain. At P38, 8 days after injection, we could no longer detect NIF via IHC for its 6xHis-tag. Because the brains are quite fragile immediately after ventrally-targeted injections (sectioning this tissue often results in a tear at the site of the injection track), it will be technically difficult to perform a comprehensive timecourse to assess NIF processing *in vivo*. However, we did perform an *ex vivo* investigation of microglia at 30 min, 60 min, and 120 min of NIF exposure, which indicated no change in NIF or CD11b levels over time (Fig. 2), suggesting that at least in *ex vivo* conditions, microglia do not change CD11b receptor levels or eliminate NIF for at least 2hrs. We have added more information on previous characterizations of NIF discussed above (lines 178-183) and specifically highlight your concern as an important caveat in the discussion (lines 208-210).

In terms of how soon D1r expression would change after NIF injection, we interpret our data not as D1rs going up, but rather D1rs failing to be eliminated via C3-microglia interactions. To clarify, we spend more time setting up the rationale for the within-animal design for these experiments (lines 196-199).

24. Figures 4 and 5 could easily be combined into a single figure.

Due to new experiments, we have retained two different behavior figures (Figs. 3,4), but each has significantly more data presented.

25. Similarly, more experiments are needed to flesh out Figure 5. What effect does NIF injection have on female social behavior? What if you inject NIF at a time point other than P30 in males? Is this effect really specific to that particular developmental timepoint or would NIF injection recapitulate this behavioral effect at any age?

Thank you for these questions. We have now injected females at P22 with NIF or Vehicle (as this is the time point prior to D1r elimination in females, Fig. 1), which can be found in Fig. 3. We have also performed NIF/Vehicle injections in both males and females at P54, a time point at which we think most of this developmental plasticity is in progress or completed. Interestingly, while we feel the data suggesting C3-microglial interactions eliminate D1rs and thus regulate social play behavior in males (P30-38) are strong, C3-microglial interactions also increase social play in females (P22-30) in a D1r-

independent fashion (Fig. 4). Manipulations at P54 in either males or females does not change social behavior, suggesting that these immune processes are mediating different outcomes in males and females, and they are confined to periods of copious development (i.e. earlier in adolescence rather than late in adolescence, when development is largely expected to be complete; Fig. S5).

26. While Figure 5 addresses the causal role of microglia in social behavior, these experiments do not demonstrate that the behavioral effect of NIF is mediated by D1 receptors. This is a key lacking piece of evidence. More in vivo experiments are needed to more tightly link each of these elements: microglia, D1 receptors, and social behavior.

We agree with this assessment, and have completed experiments using siRNA against rat D1r co-injected with either vehicle or NIF to address this concern (Fig. 4). The data indicate that NIF injection at P30 in males increases play at P38 in a D1r dependent manner (NIF + (scrambled) scRNA significantly different from all groups). Unexpectedly, NIF injection at P22 in females also increases play at P30, in a D1r independent manner. We think these data reveal very interesting sex differences in the utilization of C3 – C3R interactions. Specifically, in males, there are overt declines in D1rs and social play, both of which are mediated by C3 – C3R interactions. In females, there are no overt changes in social play, and D1rs are not eliminated by C3 – C3R mechanisms, and thus we hypothesize that C3 – C3R interactions are (i) mediating the regulation of other synaptic proteins in females and/or (ii) are engaged at a lower, more homeostatic level to regulate basal levels of synaptic proteins and thus basal levels of social play. We discuss these interesting findings in lines 329-347, and look forward to exploring this further in future experiments.

27. Generally, how specific are the observed effects (behavioral and histological) to NAc? Do D1 receptors in other brain regions show the same pattern?

This is a very good question, and one we hope to assess in future experiments. D1r autoradiography experiments suggest similar pruning or elimination in other regions of the reward circuitry (prefrontal cortex, ventral striatum) during adolescence (Andersen et al. 2000; Tarazi et al. 1999; 2000), but we have not yet assessed D1r regulation in different regions at P20/22, 30, 38, and 54 as we have done in the NAc.

Reviewer #4:

28. A main concern is that the manuscript is comprised of two distinct and only weakly related parts – the immunocytochemical/colocalization part and the in vivo behavioral part. This is particularly so because the amount of work in vivo with NIF appears to be a single experiment, in males only, and with only a vehicle control. The story developed in the first part of the paper is that there is a sex difference in loss of D1Rs. The authors need to test whether there is, is not, a sex difference in the effects of NIF. Ideally, there would a scrambled or other control for NIF. At least they could take advantage of the U-shaped NIF dose-response relationship and give higher doses of NIF in vivo which would be predicted to not prevent loss of D1Rs and not to ‘rescue’ the behavior. Also, to more rigorously interrogate their hypothesis the authors need to explore possible off-target effects by delivering NIF at times when the first part of the paper would suggest no involvement of complement (ie. P54 in males, or any time in females).

Thank you for this very thoughtful assessment of the data. We have now included NIF injections and subsequent behavioral testing in females, as well as heeded your suggestion to assess the behavioral effects of NIF in P54 males and females. To our surprise, NIF injection in females (at P22, prior to their D1r downregulation) did, in fact, slightly increase their social play (but not social exploration) behavior (Fig. 3). Consistent with our initial data indicating that NIF injection at this time did not change D1r levels in females (Fig. 2), this behavioral effect is D1r-independent (Fig. 4). These data made your suggestion of

a P54 manipulation particularly important, and we now demonstrate that P54 NIF injection does not change behavior in either males or females (Fig. S5). Collectively this report suggests that C3-microglial interactions have a sex-specific and developmental stage-specific role in D1r regulation and social play. We think the addition of the siRNA experiments and the test of age (i.e. development)-specificity in P54 manipulations now tie the paper together more cohesively with the initial D1r characterizations.

29. Even so, the first part of the paper is about D1Rs and there is no evidence (or even test) of involvement of D1Rs in the behavioral change shown in Fig. 5B. The authors need, again, to better interrogate their hypothesis, which would predict that the increase in total play time is dependent upon increased levels of D1Rs. This prediction needs to be tested experimentally. Otherwise there is only a weak, circumstantial link between first and second parts of the paper. To the extent that the NIF experiments with D1R immunofluorescence imply the blocking complement-mediated phagocytosis blocks the drop in D1Rs in males, why do D1R levels not increase again at P54 when phagocytosis seems to be lacking.

Thank you; we have now included D1r siRNA + NIF experiments to demonstrate that the effect on social play is, in fact, D1r-mediated in males, but not females (Fig. 4). Your comment regarding D1r levels increasing at P54 when there is no evidence of phagocytic activity is an interesting one. While as a field we do not understand all the nuances that guide neural development, synaptic pruning is a normal, developmentally programmed phenomenon observed in many different brain regions. While not all pruning need be through microglial phagocytosis, it certainly seems that the 'rules' of development are such that this process is integral to neural maturation, and thus a subsequent increase or rebound in receptor levels may not be advantageous. Oddly enough, new IHC data does suggest there is a trend toward increased phagocytic activity at P54 (CD68 levels) in females, but not males (Fig. 1).

30. To further support the necessity of the cells and pathways they are investigating the authors need to have more than just a single reagent, NIF. They should try interventions to eliminate or silence microglia, or animals in which key components of the pathways have been eliminated genetically. Either of these would greatly strengthen the conclusions of the paper.

We agree with the reviewer of the strengths of genetic manipulations. However, our primary behavioral output is social behavior, in particular social play behavior, which is far more robust and well-characterized in rats than it is in genetically tractable organisms like mice. We selected NIF as our primary pharmacological reagent because of its very well-characterized action, receptor specificity, and tolerance in multiple organisms, and have expanded our discussion of the history of this inhibitor in lines 178-183 (see also our response to critique #23). We do think that your suggestions, in line with your fellow reviewers, to test the necessity of D1rs for the behavioral effects of NIF has helped to illuminate the specificity of this inhibitor, as the data demonstrate sex-specific and age-specific behavioral outcomes. Specifically, NIF-induced increases in social play in males requires D1rs, while NIF-induced increases in basal levels of social play in females does not (Fig. 4). Moreover, NIF treatment in both sexes later in development (P54), did not change social behaviors (Fig. S5).

31. There are important concerns with the co-IP experiments. If D1Rs are indeed 'tagged' by iC3b then, from previous literature on this highly reactive fragment, this should form a covalent ester bond, that would not be broken even under boiling in SDS, and should see a complex of iC3b-D1R which would run at a higher molecular weight than that shown. Thus, seeing C3 immunoreactivity at the low molecular weight shown is very curious and not consistent with the authors' interpretation. They should show the entire blot to see if there are bands at the correct predicted size. Also, to fully support the conclusions the experiments should do reciprocal experiments to determine whether IPing C3 (iC3b) can co-IP D1R. In addition, the authors present no evidence for quantification of the

results shown in the images, nor whether the images are representative of replicates or single experiments. Finally, they need to provide a credible explanation for the high iC3b signal in the colIP at P54F, when both the D1R and C3 levels are at their lowest.

Thank you for your thorough critique. Hydroxymethylamine incubation is often used to break ester bonds and release iC3b or other covalently bonded C3 fragments. Because of the very low volume levels needed to resolve proteins in our Co-IP assay, we did not want to add hydroxymethylamine in addition to the normal lysis buffer we use for fear of losing our signal, but we do always perform our Western blotting by boiling samples in a buffer containing beta-mercaptoethanol (BME). BME has also been demonstrated to break ester bonds (thioester bonds in particular; Bolanowski et al. 1984), and is a strong nucleophile, like hydroxymethylamine. We thus felt comfortable focusing on the ~42kDa band, as it is the easiest of the iC3b-specific bands to resolve and quantify. However, your expertise has helped us reflect on the goals of the current manuscript, and we have decided to leave the nature of the C3 tag for discussion (lines 371-386) rather than provide the current circumstantial data. Co-IPs, while better than typical immunofluorescence, are still correlational, and an in-tissue variation of FRET or proximity ligase amplification would be much more conclusive. We look forward to fleshing out this line of inquiry in future experiments.

Other concerns

32. There is substantial inconsistency in the images in the panels in Fig 1. As one example, in panel B P30 the maximum C3 volume should be about 0.5% whereas the average for what should be shown in panel F P38 should be about 0.4%. Yet the image in B is massively brighter than that in F. Likewise, C P30 should have only about 20% more immunofluorescence than G P20, which is not consistent with the images. This dramatic inconsistency undercuts the reliability of the data shown in this and all immunofluorescence figures.

Thank you; importantly, male and female tissue needed to be imaged with distinct acquisition parameters (i.e. relative laser strength), and should not be directly compared. Data can only be directly compared across ages in each individual sex. We have re-checked to ensure our representative images represent the average data in both males and females, and have made this important caveat more clear in the text and figure legends (lines 132-134; 791-793).

33. The authors regularly refer to phagocytic elimination of D1Rs. What do they mean by this – that the receptors are somehow plucked from the neuronal membrane, that substantial chunks are taken from the neurons, or something else?

This is an excellent question. There is not yet consensus on how microglial phagocytosis of synaptic elements occurs. There was recently an excellent report showing for the first time microglial synaptic pruning, and the authors note that microglia participate in “trogocytosis,” or synaptic nibbling, in which small synaptic elements can be sampled and removed (Weinhard et al. 2018). We have added more information about this recently released report (lines 94-100). Herein, we remain agnostic to the extent to which microglia are eliminating D1r-containing synapses (i.e. trogocytosis vs. full phagocytosis).

34. The authors find that NIF doesn't completely eliminate phagocytosis. It would be interesting to compare effect of NIF to that of Cd11b KO to determine how much of the Cd11b-dependent phagocytosis is blocked.

We agree, this will be a very interesting set of future experiments. This said, we believe the results of this type of experiment would need to be interpreted with caution, as a CD11b KO would have additional, brain and systemic effects (and would not be brain region specific). Thus, we believe the NIF manipulation is far more specific for testing our hypothesis.

35. The other measures the authors saw correlate with D1R loss (increased C3-D1R microglia contact and/or increase D1R in CD68 lysosomes) should be examined with NIF as further confirmation that the drug is acting on this phagocytosis pathway.

We agree this is important, and we think that the new behavioral data indirectly shed light on this comment. When we decrease D1r expression via siRNA while also providing NIF (which increases D1r levels through inhibition of microglia-C3 interactions), the behavioral effect of NIF is abrogated in males (Fig. 4). Thus downregulating D1r through alternative mechanisms was able to compensate for the effects of NIF.

36. A point which seems to be ignored by the authors is that the D1R volume in males is much lower than that in females. What is the explanation for that in the context of the present study?

Thank you for this comment. We cannot directly compare D1r levels between the sexes for the reasons discussed above, and have made this more clear in the text in lines 132-134; 791-793. Please also see our response to critique #4 for a more explicit examination of the representative images.

37. Given the NIF dose response in Fig 3d, how did the authors select the dose to be used in vivo, given that the NIF is differentially diluted when injected in vivo as compared with when applied in the bathing solution in vitro?

While the dilution is different, we used the same total number of ng for each experiment (60ng). The final concentration of drug to be tested was constrained by the volume that could be infused into the NAc *in vivo*, which happened to be a maximum of 300nL in P30 males (250nL in females). We then reconstituted NIF at either the recommended reconstitution concentration (200µg/mL) or 2x this (400µg/mL). 300nL at each of those concentrations is a total of 60ng and 120ng, respectively.

38. In many places in the manuscript the writing is excessively leading, presuming the conclusions before doing the experiments.

Thank you for this comment; we try to be clear about our predictions, interpretations of the data, and what we view is the next logical question, but have gone back through the manuscript and tried to be more conservative in our writing.

Reviewers' comments:

Reviewer #1 (Remarks to the Author):

The revision by Kopec et al. adds a large amount of new data that address most concerns of the reviewers. However, one major concern remains.

The authors have provided data to address the concern of antibody specificity by showing that D1R antibodies overlap with D1 MSNs (via a td-Tomato tag). While these data show that their antibody overlaps with cells that express D1 receptors, this is not a sufficient validation for a manuscript that relies almost entirely on the validity of these antibodies. Td-Tomato expressed on the D1 promoter only shows that the antibody labels proteins in cells that also express D1 receptors, however, the tag is not the receptor itself. Second, with non-specific antibodies, they often do label the proteins that they are said to. The problem most often is that they also bind to other targets - a great example is the kappa opioid receptor antibody, which will label the purified kappa receptor on a western blot but will also show expression in a kappa receptor knockout mice. Thus, the appropriate way to validate this antibody is to show lack of expression in a D1 receptor. This will validate that there is not off-target binding, which is critical for the conclusions being drawn in the current study.

Reviewer #2 (Remarks to the Author):

The authors have extensively revised the manuscript and performed additional behavioral analyses, following the reviewers suggestions. In my opinion, the paper contains novel information that will be of interest for the readers. The revised version is suitable for publication

Reviewer #3 (Remarks to the Author):

The authors have addressed most of my comments with one major exception—comment #16: validation of their D1 receptor antibody specificity is still inadequate (Figure S1A). They only show at very low magnification that D1R is expressed in roughly similar regions as TdTomato in a transgenic D1-TdTomato mouse. Since D1R and D2R are expressed in largely overlapping regions, the data as shown do not exclude the possibility of off-target labeling of D2Rs (or other proteins). What they should have provided is a higher magnification cellular resolution quantification of % overlap between TdTomato-expressing and D1R antibody-labeled cells. Of particular interest/concern is the number of D1R antibody-stained cells not expressing TdTomato, which is not discussed or quantified. The results of the entire paper rest on the assumption of specificity of this antibody for D1 receptors, so this is a very important validation that is lacking.

Regarding new experiments on in vivo NIF infusions (reference to comments #23 and #26): they are indeed addressing the causal role of D1 receptors in the NIF-mediated behavior effects, which strengthens the paper. Why are the D1R expression data only presented as

% of vehicle-treated hemisphere? How does this data compare to the raw numbers in Figure 1? Does vehicle infusion alone affect D1R immunoreactivity? N = 4 animals is a relatively small sample size. The additional information on NIF action and time course in the text is appreciated, although the authors acknowledge there is no precedent of NIF infusion directly into brain, leaving some of my questions unanswered. I don't understand the argument as to why an in vivo time course is not possible because the tissue tears upon sectioning—it is not clear why a single stereotaxic drug infusion with a syringe would affect tissue slicing even 24 hours later (this would be something I would expect with an implanted cannula or very large gauge needle, for example).

The mixed results observed in females presented in parallel with the male data throughout the paper are confusing and seem to raise more questions than they answer—complicating the story and leaving many untied ends. These somewhat confusing sex differences aside, overall the writing of the manuscript has improved and provides important clarification without overstating the findings.

Reviewer #4 (Remarks to the Author):

With the revisions to the text and figures, and comments to some of my concerns, I find that the manuscript is acceptable for publication.

Response to Reviewers

We thank reviewers #2 and #4 for their support of the manuscript as well as reviewers #1 and 3 for their comments and constructive critique. Below we respond point-for-point to the reviewers' critiques.

Reviewer #1

1. The revision by Kopec et al. adds a large amount of new data that address most concerns of the reviewers. However, one major concern remains. The authors have provided data to address the concern of antibody specificity by showing that D1R antibodies overlap with D1 MSNs (via a td-Tomato tag). While these data show that their antibody overlaps with cells that express D1 receptors, this is not a sufficient validation for a manuscript that relies almost entirely on the validity of these antibodies. Td-Tomato expressed on the D1 promotor only shows that the antibody labels proteins in cells that also express D1 receptors, however, the tag is not the receptor itself. Second, with non-specific antibodies, they often do label the proteins that they are said to. The problem most often is that they also bind to other targets - a great example is the kappa opioid receptor antibody, which will label the purified kappa receptor on a western blot but will also show expression in a kappa receptor knockout mice. Thus, the appropriate way to validate this antibody is to show lack of expression in a D1 receptor. This will validate that there is not off-target binding, which is critical for the conclusions being drawn in the current study.

Thank you for this comment. The Caron lab at Duke University has generously donated D1r KO brains and wildtype controls, and we now demonstrate a complete absence of anti-D1r immunoreactivity in D1r KO brains with clear staining in WT brain, demonstrating the specificity of the antibody (Fig. S1B).

Reviewer #2

2. The authors have extensively revised the manuscript and performed additional behavioral analyses, following the reviewers suggestions. In my opinion, the paper contains novel information that will be of interest for the readers. The revised version is suitable for publication

Reviewer #3

3. The authors have addressed most of my comments with one major exception—comment #16: validation of their D1 receptor antibody specificity is still inadequate (Figure S1A). They only show at very low magnification that D1R is expressed in roughly similar regions as TdTomato in a transgenic D1-TdTomato mouse. Since D1R and D2R are expressed in largely overlapping regions, the data as shown do not exclude the possibility of off-target labeling of D2Rs (or other proteins). What they should have provided is a higher magnification cellular resolution quantification of % overlap between TdTomato-expressing and D1R antibody-labeled cells. Of particular interest/concern is the number of D1R antibody-stained cells not expressing TdTomato, which is not discussed or quantified. The results of the entire paper rest on the assumption of specificity of this antibody for D1 receptors, so this is a very important validation that is lacking.

Thank you for these comments. The Caron lab at Duke University has generously donated D1r KO brains and wildtype controls, and we now demonstrate an absence of anti-D1r immunoreactivity in D1r KO brains, demonstrating the specificity of the antibody (Fig. S1B). We have retained the TdTomato images at lower magnification to demonstrate the spatial pattern of overlap in D1r-TdTomato and anti-D1r antibody immunoreactivity, and now add a higher magnification (60x confocal microscopy as in Fig. 1) to demonstrate similar anti-D1r staining patterns in wildtype, but not D1r KO mouse striatum.

4. Regarding new experiments on in vivo NIF infusions (reference to comments #23 and #26): they are indeed addressing the causal role of D1 receptors in the NIF-mediated behavior effects, which

strengthens the paper. Why are the D1R expression data only presented as % of vehicle-treated hemisphere? How does this data compare to the raw numbers in Figure 1?

When we expressed these data as the raw densitometric values, we sensed some confusion over the magnitude of the NIF effect relative to within-animal vehicle controls in the first set of reviews, likely because there was variability in basal D1r levels. Now expressing the NIF-manipulated data as a percentage of change from within-animal vehicle-manipulated hemispheres, the magnitude of NIF-induced changes (~25-30% increase in D1r immunoreactivity) is clearer. We will retain these figures in the main manuscript (Fig. 2F,H), but have also now included the original raw data in Supp. Fig. 2F, G. The image acquisition and analysis of these data were slightly different from those in Fig. 1, based on our goals. In Fig. 1 we were most interested in the relationships between proteins and cell types, so we acquired 60x magnification confocal z-stacks with 0.1 μ m steps for Imaris 3D reconstructions, which ultimately provided detailed volumetric measurements. For NIF effects on D1r immunoreactivity, we wanted to assess more global changes in D1r levels in the NAc, and thus used lower magnification (10x) epifluorescent z-stacks with 0.5 μ m steps. These z-stacks were processed into an average projection and densitometric measurements were acquired with ImageJ. For these reasons, we cannot directly compare data between Fig. 1 and Fig. 2.

5. Does vehicle infusion alone affect D1R immunoreactivity? N = 4 animals is a relatively small sample size.

The question of whether vehicle infusion alone affects D1r immunoreactivity is an interesting one. The reason we did not assess this is because we never compared a non-surgically manipulated animal to a surgically manipulated animal. We agree that surgical manipulations alone can cause changes in molecular and behavioral outcomes, which is why it is important to only draw comparisons between animals with the most comparable experiences (save for the independent variable) possible. The vehicle in these experiments is sterile PBS, and we utilized a within-animal control design for assessing the impact of NIF on D1r immunoreactivity to best control for the animal's prior experiences and its response to surgical intervention. Regarding the sample size, we performed a post-hoc analysis of achieved power given the mean and standard deviation of the differences from this data set. The effect size for our data (and this particular statistical test, the one-sample t-test), is 2.07, which, using G*Power software, results in a power of 0.92. A well-powered experiment is typically 0.8-0.9. Together, the large effect size and power indicate that despite a low n , this is a well-powered design with a low likelihood of false positives ($\alpha=5\%$) or false negatives ($\beta=8\%$).

6. The additional information on NIF action and time course in the text is appreciated, although the authors acknowledge there is no precedent of NIF infusion directly into brain, leaving some of my questions unanswered. I don't understand the argument as to why an in vivo time course is not possible because the tissue tears upon sectioning—it is not clear why a single stereotaxic drug infusion with a syringe would affect tissue slicing even 24 hours later (this would be something I would expect with an implanted cannula or very large gauge needle, for example).

Thank you for clarifying, we may have misinterpreted your original request for a timecourse. Hours after injection, the tissue is still very sensitive and difficult to cryosection. Starting after 24 hours, tissue is more manageable. We originally thought you were asking for an *in vivo* time course on the range of hours, which is why we completed an *ex vivo* time course on the range of hours. We have now included an assessment of *in vivo* NIF immunoreactivity on the range of days, which indicates the NIF immunoreactivity can be detected for 3-5 days after stereotaxic microinjection (Fig. S2E).

7. The mixed results observed in females presented in parallel with the male data throughout the paper are confusing and seem to raise more questions than they answer—complicating the story and

leaving many untied ends. These somewhat confusing sex differences aside, overall the writing of the manuscript has improved and provides important clarification without overstating the findings.

We appreciate that the story is complex. While it is beyond the scope of this manuscript to determine the mechanisms underlying the striking sex differences in the endpoints we have described, we strongly believe that the inclusion of both sexes is important and should be retained, as it will lay the foundation for exploring mechanisms in future experiments, by us and by others.

Reviewer #4

8. With the revisions to the text and figures, and comments to some of my concerns, I find that the manuscript is acceptable for publication.

REVIEWERS' COMMENTS:

Reviewer #1 (Remarks to the Author):

The authors have done an excellent job responding to the comments. I have no further comments on the manuscript.

Response to Reviewers

We thank both reviewers for their support of the manuscript. Below we respond to the Reviewer #2's critique.

Reviewer #1

The authors have done an excellent job responding to the comments. I have no further comments on the manuscript.

Reviewer #2 (paraphrased from editor)

"...reviewer #2 still suggests that you should quantify the percent of antibody labeled cells that colocalize with D1-tdTomato, as suggested in the last round of review. We ask that you provide this quantification unless you have a strong reasoning as to why this is not appropriate..."

Thank you for these comments. The addition of the D1r KO data, where there is no signal resulting from the D1r antibody in the KO brain, but characteristic striatal signaling in the wildtype control brain, as suggested by reviewers, is the most appropriate antibody validation control. We retained the tdTomato image in the manuscript to be comprehensive, and also to highlight our gratitude to both the Lobo and Caron labs which generously shared brain tissue with us to complete these analyses. Unfortunately, as this tissue was a gift and not the primary model in this manuscript, we have only an $n=1$, making quantification difficult to interpret. If in future experiments we use the D1r-tdTomato mouse model more extensively, we agree that this will be an important and informative analysis to perform.